# Whole-Song Hierarchical Generation of Symbolic Music Using Cascaded Diffusion Models

**Ziyu Wang[12], Lejun Min[2], Gus Xia[21]**
[1]Computer Science Department, NYU Shanghai, [2]Machine Learning Department, MBZUAI
`ziyu.wang@nyu.edu`, {`lejun.min, gus.xia`}`@mbzuai.ac.ae`

## Abstract

Recent deep music generation studies have put much emphasis on long-term generation with structures. However, we are yet to see high-quality, well-structured **whole-song** generation. In this paper, we make the first attempt to model a full music piece under the realization of *compositional hierarchy*. With a focus on symbolic representations of pop songs, we define a hierarchical language, in which each level of hierarchy focuses on the semantics and context dependency at a certain music scope. The high-level languages reveal whole-song form, phrase, and cadence, whereas the low-level languages focus on notes, chords, and their local patterns. A cascaded diffusion model is trained to model the hierarchical language, where each level is conditioned on its upper levels. Experiments and analysis show that our model is capable of generating full-piece music with recognizable global verse-chorus structure and cadences, and the music quality is higher than the baselines. Additionally, we show that the proposed model is *controllable* in a flexible way. By sampling from the interpretable hierarchical languages or adjusting pre-trained external representations, users can control the music flow via various features such as phrase harmonic structures, rhythmic patterns, and accompaniment texture.[1]

## 1 Introduction

In recent years, we have witnessed a lot of progress in the field of deep music generation. With significant improvements on the quality of generated music (Copet et al., 2023; Thickstun et al., 2023) on short segments (typically ranging from a measure up to a phrase), researchers start to put more emphasis on *long-term structure* as well as how to *control* the generation process in a musical way. The current mainstream approach of structural generation involves first learning disentangled latent representations and then constructing a predictive model that can be controlled by the learned representations or external labels (Yang et al., 2019; Wang et al., 2020b; Wei et al., 2022; Chen et al., 2020). However, generating an entire song remains an unresolved challenge. As compositions extend in length, the number of involved music representations and their combinations grow exponentially, and therefore it is crucial to organize various music representations in a structured way.

We argue that *compositional hierarchy* of music is the key to the solution. In this study, we focus on symbolic pop songs, proposing a computational *hierarchical music language* and modeling such language with cascaded diffusion models. The proposed music language has four levels. The top-level language describes the phrase structure and key progression of the piece. The second-level language reveals music development using a reduction of the melody and a rough chord progression, focusing on the music flow within phrases. The third-level language consists of the complete lead melody and the finalized chord progression, which is usually known as a lead sheet, further detailing the local music flow. At the last level, the language is defined as piano accompaniment. Intuitively, the language aims to characterize the intrinsic homophonic and tonal features of most pop songs—a verse-chorus form, a chord-driven tonal music flow, and a homophonic accompaniment texture.

---

[1]We release the complete source code and model checkpoints at `https://github.com/ZZWaang/whole-song-gen`. The demo page is available at `https://wholesonggen.github.io`.

We represent all levels of the symbolic languages as multi-channel images and train four layers of image diffusion models in a cascaded fashion, one for each level of the music language. The generation scope of the first layer is full-song and up to 256 measures, the scope of the second layer is 32 measures, and the third and the fourth layer each has a scope of 8 measures. Additional autoregressive controls are added to the low-level diffusion models to strengthen long-term temporal coherency. Experimental results show that our model is capable of generating well-structured full-piece music with recognizable verse-chorus structure and high music quality.

Moreover, at each level, optional *external conditions* can be added via the cross-attention mechanism of diffusion models to control the generation process at each level of the hierarchy. As a demonstration, we add long-term control of chord progression, local control of rhythmic and accompaniment pattern to the corresponding levels of the hierarchy. All the external controls uses pre-trained latent codes from existing music representation learning models. We show that these controls can effectively guide hierarchical generation in a more customizable way.

In summary, the contribution of the paper is as follows:

- **We achieve high-quality and well-structured whole-song generation** with cascaded diffusion models. Objective and subjective measurement show that both monophonic lead sheets and polyphonic accompaniment generated by our model have more identifiable phrase boundaries, better-structured phrase development in similarity and contrast, and higher music quality compared to baselines.

- **We propose a computational hierarchical music language** as a structural inductive bias, making the training process decomposable and efficient in terms of data and computing power utilization. Also, the hierarchical languages can be extracted automatically without manual annotation of music structure.

- **Our model enables flexible and interpretable controls**, with not only our proposed hierarchical language but also with external pre-trained latent representations, such as chord, melodic rhythm and accompaniment texture.

## 2 RELATED WORK

In this section, we first review music structure in musicology in Section 2.1, followed by music structure modeling in deep music generation approaches in Section 2.2. Finally, in Section 2.3 we review the state-of-the-art deep generative methods relevant to the problem of whole-song generation.

### 2.1 MUSIC STRUCTURE MODELING

Traditional music theory focuses on the analysis of music structure in terms of counterpoint (Clementi et al., 2010), harmony (Schoenberg, 1983), forms (Koch, 1787), etc. In the early 20th century, a more comprehensive theory, *Schenkerian analysis* (Schenker, 1979), emerged with a focus on the *generative* procedure of music. The theory introduces a compositional hierarchy of music, aiming to show how a piece is composed from its *background*, the normal form of music, to its *middle ground*, where music form and rough music development are realized, and finally to the *foreground*, the actual composition.

Nowadays, compositional hierarchy is still prevalent in modern musicology. Notable developments include Tagg (1982), a general compositional hierarchy for pop music, and *Generative Theory of Tonal Music* (GTTM) (Lerdahl & Jackendoff, 1996), a theory focusing on the definition and analysis of formal musical syntax. From a computational perspective, these studies provide more formal music features and computer-friendly generative processes (Hamanaka et al., 2015; 2016). The focus of this paper is to further leverage the compositional hierarchy of music to develop a fully computable language and to model it with deep neural networks.

### 2.2 STRUCTURED DEEP MUSIC GENERATION

Recent advances in deep generative models have greatly improved music generation quality, primarily by more effective modeling of the local musical structure in two ways: implicit and explicit.

Implicit approaches, exemplified by models such as Music Transformer (Huang et al., 2019), Muse-BERT (Wang & Xia, 2021), and Jukebox (Dhariwal et al., 2020), learn structures by predicting and filling musical events, often revealing context dependencies via attention weights. Explicit approaches leverage domain knowledge to define music features or extract interpretable music representations, allowing the learning of structures like measure-level pitch contour and accompaniment (Yang et al., 2019; Wang et al., 2020b; Dai et al., 2021; Zhao et al., 2023). This study aims to combine both explicit and implicit approaches and further model phrase and whole-song structures. The explicit modeling lies in our definition of a computational hierarchical music language, and the implicit modeling of the structure lies in the cascaded diffusion models.

## 2.3 Diffusion and Cascaded Modeling for Music Generation

Diffusion models, after their success in image and audio domains, have very recently been applied to music generation (Mittal et al., 2021; Li & Sung, 2023; Min et al., 2023). Besides high sample quality, diffusion models naturally lead to coherent local structures with the innate inpainting method Lugmayr et al. (2022), i.e., by generating music segments conditioned on surrounding contexts. As for long-term structures, we recently saw the design of cascaded diffusion modeling in Moûsai Schneider et al. (2023), which generates high-fidelity audios using multi-scale sampling.

In this study, our focus is on symbolic music and we adopt the idea of multi-scale generation with cascaded models. Additionally, we integrate the cascaded process with the proposed hierarchical music language, so that each layer of the diffusion model focuses on a certain interpretable aspect of music composition. In particular, all levels of music languages are defined as image-like representations. Inspired by sketch- and stroke-based image synthesis Cheng et al. (2023), we model hierarchical music generation by regarding high-level and low-level music languages as the background and foreground "strokes", respectively.

## 3 Methodology

Our model for whole-song generation is a realization of the music compositional hierarchy. In this section, we first introduce the definition of our hierarchical music languages in Section 3.1. Then, we discuss how to model these languages via cascaded diffusion models, where each level of the language is conditioned on its upper levels. We show the data representation of these languages in Section 3.2. The training and inference of the model are discussed in Section 3.3 and Section 3.4, respectively.

### 3.1 Definition of Hierarchical Music Languages

We define a hierarchical music language with four levels to reveal the generative procedure of music, as shown in Table 1. The highest level, *Form*, includes music keys and phrases. This is followed by *Reduced Lead Sheet*, which contains reduced melody and simplified chords. The third level, *Lead Sheet*, includes the lead melody and chords. The final level, *Accompaniment*, consists of the piano accompaniment. The key idea behind this hierarchical design lies in the relationship among the four levels—more abstract music concepts at higher levels are realized by stylistic specifications at lower levels. For example, a lead sheet is an abstraction implying many possible ways to arrange the accompaniment that share the same melodic and harmonic structure, while an instantiated accompaniment is one of the possible realizations showing the accompaniment structure in more detail.

Note that for Form, Lead Sheet and Accompaniment, there are established music information retrieval algorithms for labeling. In contrast, the Reduced Lead Sheet is a unique design, which we refer the readers to Appendix A for details.

### 3.2 Data Representation

While music scores are inherently symbolic, we transform them into continuous, image-like piano-roll representations for better compatibility with diffusion models. Specifically, languages at all levels are represented by multi-channel images (examples are shown in Appendix A). The image width represents sequence length under different resolutions, and the height represents 128 MIDI

Table 1: Definition of the four-level hierarchical music language. We use m for measure, b for beat, s for step, to represent the temporal resolution. $M$ denotes the number of measures in a piece, $\gamma$ denote the number of beats in a measure, and $\delta$ denotes the number of steps in a beat.

| Languages (res.) | Specification | Data Representation | Structural Focus |
|---|---|---|---|
| Form (m) | Key changes Phrases | $\boldsymbol{X}^1 \in \mathbb{R}^{8 \times M \times 12}$ | Music form |
| Reduced Lead Sheet (b) | Melody reduction Simplified chord | $\boldsymbol{X}^2 \in \mathbb{R}^{2 \times \gamma M \times 128}$ | Phrase similarity, phrase development & cadence |
| Lead Sheet (s) | Lead melody Chord | $\boldsymbol{X}^3 \in \mathbb{R}^{2 \times \delta \gamma M \times 128}$ | Melodic pattern, similarity & coherence |
| Accompaniment (s) | Accompaniment | $\boldsymbol{X}^4 \in \mathbb{R}^{2 \times \delta \gamma M \times 128}$ | Acc. pattern, similarity & coherence, Mel-acc relations |

pitches or 12 pitch classes. We denote the piece length to be $M$ measures, each measure containing $\gamma$ beats, and each beat containing $\delta$ steps. In this paper, we consider $\gamma \in \{3, 4\}$ and $\delta = 4$.

The language Form is a sequence of keys and phrases under the resolution of one measure. Keys are represented by $\boldsymbol{K} \in \mathbb{R}^{2 \times M \times 12}$, where tonic information and scale information are stored on the two channels with binary values. For phrases, we use $\boldsymbol{P} \in \mathbb{R}^{6 \times M \times 1}$, where six channels correspond to six phrase types (e.g., verse and chorus, see Table 3 for more detail), and the pixel values indicate measure countdown. Formally, let $m_0, ..., m_0 + L - 1$ be the indices of a $L$-measure phrase of type $i_0$, then for $m_0 \le m < m_0 + L$,

$$\boldsymbol{P}[i, m, :] := \mathbb{1}_{\{i=i_0\}}(1 - \frac{m - m_0}{L}). \tag{1}$$

We broadcast $\boldsymbol{P}$ to match the pitch-axis of $\boldsymbol{K}$ and define the first-level language Form as $\boldsymbol{X}^1 := \text{concat}(\boldsymbol{K}, \boldsymbol{P}) \in \mathbb{R}^{8 \times M \times 12}$. The other levels of languages use a piano-roll representation. Reduced Lead Sheet is represented by $\boldsymbol{X}^2 \in \mathbb{R}^{2 \times \gamma M \times 128}$ under the resolution of one beat, where two channels correspond to note onset and sustain. Both melody reduction and simplified chord progression share the same piano-roll using different pitch registers. Similarly, Lead Sheet uses $\boldsymbol{X}^3 \in \mathbb{R}^{2 \times \delta \gamma M \times 128}$ to represent the actual melody and chords, and Accompaniment uses $\boldsymbol{X}^4 \in \mathbb{R}^{2 \times \delta \gamma M \times 128}$ to represent the accompaniment, both in the same resolution of one step.

Note that for the four levels $k = 1, \ldots, 4$, $\boldsymbol{X}^k$ have different shapes. In the following sections, we write $\{\boldsymbol{X}^k | k \subset \{1, 2, 3, 4\}\}$ to denote the concatenation along the channel axes with possible broadcasting and repetition operations. For example, $\boldsymbol{X}^1$ can be expanded $\gamma$ times in width and repeated 11 times in height to be concatenated with $\boldsymbol{X}^2$, resulting in a tensor $\boldsymbol{X}^{\le 2} \in \mathbb{R}^{10 \times \gamma M \times 128}$. Additionally, we write the time-series expression $\boldsymbol{X}^k_t$ to denote $\boldsymbol{X}^k[:, t, :]$ for simplicity.

### 3.3 MODEL ARCHITECTURE

Whole-song music generation is achieved by generating the four levels of hierarchical music languages one after another in a top-down order (as shown in Figure 1). For each level, we train a diffusion model to realize the current-level language based on the existing upper-level languages. The *actual scopes* (image widths) of these diffusion models are generally the same, yet the *music scopes* vary significantly since the resolution in lower-level languages is finer. In this paper, for level $k = 1, ..., 4$, we set the actual scope $b_k$ to be $b_1 = 256$ and $b_{2:4} = 128$, which means the music scopes for these levels are 256 measures, 128 beats, 128 steps, and 128 steps, respectively. In the usual setting when $\gamma = \delta = 4$, the music scopes of the models are 256 measures, 32 measures, 8 measures, and 8 measures, respectively. Consequently, except that the first layer is an unconditional generation of the whole sequence, the generation at all the other layers are essentially conditional generation of music segments sliced from the entire sequences.

The generation of a music language slice $\boldsymbol{X}^k_{t:t+b_k}$ at level $k \ne 1$ can be conditioned on multiple resources inside and outside the defined hierarchy. In this study, our model is designed to take in three sources of structural conditions:

**Background condition.** We regard the generation as a realization of existing higher-level languages at the corresponding scope $\boldsymbol{X}^{<k}_{t:t+b_k}$, where the higher-level language segments are like sketch images

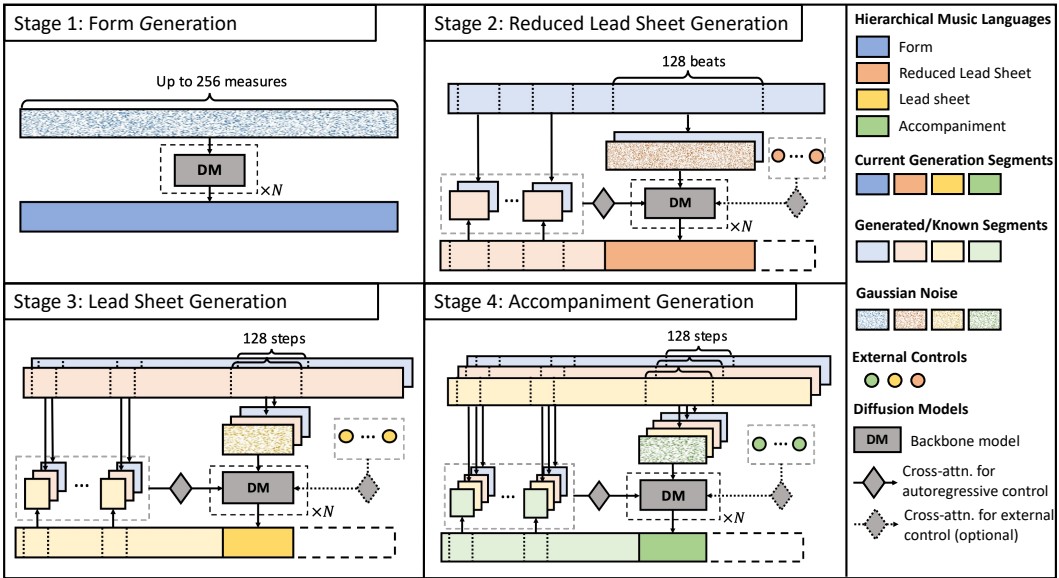

Figure 1: The diagram of cascaded diffusion models for hierarchical symbolic music generation.

directly guiding the current generation. Background condition is applied by concatenating the input with $\boldsymbol{X}^{<k}_{t:t+b_k}$ along the channel axis.

**Autoregressive condition.** The segment should not only be a realization of the background condition, but also coherent with prior realizations $\boldsymbol{X}^{\leq k}_{<t}$. For example, the realization of a verse phrase in the end of the composition is usually similar to the realization in the beginning. In our model, we make an autoregressive assumption that $\boldsymbol{X}^k_{<t}$ are known. We select $S_k$ relevant music segments prior to $t$ based on a defined similarity metric on $\boldsymbol{X}^1$. These music segments are encoded into latent representations and being cross-attended in the diffusion models.

**External condition.** Besides the compositional hierarchy, music generation can be controlled by other external conditions. These conditions can be stylistic controls of multiple scopes (Wei & Xia, 2021; Yang et al., 2019; Wang et al., 2020b), or cross-modality controls of text (Zhang et al., 2020) or audio (Wang et al., 2022). As an illustration of our model compatibility, we use pre-trained latent representations of chords, rhythmic pattern, and accompaniment texture as the control for Reduced Lead Sheet, Lead Sheet, and Accompaniment generation, respectively. At each level $k$, we denote the array of external latent codes by $\boldsymbol{Z}^k_{t:t+b_k}$, which are cross-attended in our diffusion models.

For all four levels, we adopt a 2D-UNet with cross-attention similar to Min et al. (2023) as the backbone neural architecture and make several modifications. First, the input channels are increased to allow background condition. Second, autoregressive and external conditions are fed through cross-attention layers with classifier-free guidance (Baykal et al., 2023). In Appendix B we include more details on the model architecture and training. Mathematically, we use diffusion to model the conditional probability of multiple levels of music segments. Let the backbone model at level $k$ be denoted by

$$\epsilon_{\theta_k}(x_n, n, y^{\text{bg}}, y^{\text{ar}}, y^{\text{ext}}), \tag{2}$$

where $\theta_k$ is the model parameter, $n = 0, ..., N$ is the diffusion step, $x_n$ is the input image mixed with Gaussian noise at diffusion step $n$, and $y^{\text{bg}}$, $y^{\text{ar}}$, and $y^{\text{ext}}$ are background, autoregressive, and external control, respectively. Our training objective is to model the probability

$$p_{\theta_k}(\boldsymbol{X}^k_{t:t+b_k} | \boldsymbol{X}^{<k}_{t:t+b_k}, \boldsymbol{X}^{\leq k}_{<t}, \boldsymbol{Z}^k_{t:t+b_k}) \tag{3}$$

under the loss function

$$\mathcal{L}(\theta_k) = \mathop{\mathbb{E}}_{\boldsymbol{X},t} \ell_{\theta_k}(\boldsymbol{X}^k_{t:t+b_k}, \boldsymbol{X}^{<k}_{t:t+b_k}, \boldsymbol{X}^{\leq k}_{<t}, \boldsymbol{Z}^k_{t:t+b_k}), \tag{4}$$

where

$$\ell_{\theta_k}(x, y^{\text{bg}}, y^{\text{ar}}, y^{\text{ext}}) = \mathbb{E}_{\epsilon,n}||\epsilon - \epsilon_{\theta_k}(x_n, n, y^{\text{bg}}, y^{\text{ar}}, y^{\text{ext}})||^2_2. \tag{5}$$

## 3.4 Whole-Song Generation Algorithm

At the inference stage, we leverage the conditional probability (see Equation 3) to achieve whole-song generation by autoregressively inpainting the generated segments. Inpainting is a commonly-used method in diffusion models for image editing, and is developed as a quasi-autoregressive method for sequential generation (Min et al., 2023). In our algorithm, we use a hop length of $h_k := b_k /\!/ 2$ for inpainting, and the algorithm is shown in Algorithm 1. Note that in training, we zero-pad $\boldsymbol{X}^1$ to 256 measures, so during inference, we derive the actual song length by finding the first all-zero entries of the generated $\boldsymbol{X}^1$. This process is denoted by INFERSONGLENGTH$(\cdot)$. Here, we use

$$\boldsymbol{X}_{t+h_k:t+b_k}^k \sim p_{\theta_0}(\cdot|\boldsymbol{X}_{t:t+h_k}^k; \boldsymbol{X}_{t:t+b_k}^{<k}, \boldsymbol{X}_{<t}^{\leq k}, \boldsymbol{Z}_{t:t+b_k}^k) \tag{6}$$

to indicate the distribution of the second half of the sequence conditioned on the first half via inpainting, together with the background, autoregressive, and external conditions.

---

**Algorithm 1** Whole-song generation algorithm.

---

**Constants**: Resolution factor for each level $r_1 = 1, r_2 = \gamma, r_3 = r_4 = \delta\gamma$
**Input**: External control $\boldsymbol{Z}^k (2 \leq k \leq 4)$ (optional)
1: $\boldsymbol{X}^1 \sim p_{\theta_1}(\cdot|\emptyset, \emptyset, \emptyset)$
2: $M \leftarrow$ INFERSONGLENGTH$(\boldsymbol{X}^1)$
3: **for** $k = 2, \ldots, 4$ **do**
4: $\quad \boldsymbol{X}_{0:h_k}^k \sim p_{\theta_k}(\cdot|\boldsymbol{X}_{0:b_k}^{<k}, \emptyset, \boldsymbol{Z}_{0:b_k}^k)$
5: $\quad$ **for** $t = 0, h_k, 2h_k, \ldots, r_k M - b_k$ **do**
6: $\quad\quad \boldsymbol{X}_{t+h_k:t+b_k}^k \sim p_{\theta_k}(\cdot|\boldsymbol{X}_{t:t+h_k}^k; \boldsymbol{X}_{t:t+b_k}^{<k}, \boldsymbol{X}_{<t}^{\leq k}, \boldsymbol{Z}_{t:t+b_k}^k)$
7: $\quad$ **end for**
8: **end for**
9: **return** $\{\boldsymbol{X}^k|1 \leq k \leq 4\}$

---

## 4 Analysis of Structural Music Generation

In this section, we show an example of whole-song music generation of **40 measures** in Figure 2. The given Form of the piece has a simple verse-chorus structure with 4-measure verse and 8-measure chorus phrases appearing multiple times.

The generated music shows a clear music structure. The melodies of three verses all consist of syncopated rhythm in a narrow pitch range, while the melodies of two choruses are both relatively lyrical with a broader pitch range (indicated by shaded rectangles). The accompaniment pattern predominantly features eighth notes in verses and sixteenth notes in choruses. Moreover, the cadences at phrase boundaries are clearly indicated by the tonic or dominant chords and the "fill" in the accompaniment (indicated by dotted red rectangles). Furthermore, we notice the music intensity in the second half is stronger than in the first half, realized by more active pitch and rhythm movements and higher pitches (indicated by shaded rectangles with dotted borders). Such intensity changes make the composition go to a climax point before ending, showing a well-formed chronological structure.

In Appendix C, we break down the hierarchical generation process and show examples of structural controllability of each level. More generation results are available at the demo page.

## 5 Experiments

We focus our experiments on the generation of Lead Sheet and Accompaniment, the two lower levels of languages. The rationale is that the information of higher-level languages is difficult to evaluate directly, and they are implied at the lower levels. In this section, we first evaluate the structure of full pieces based on a proposed objective metric, and then subjectively evaluate both structure and quality on music segments (8-measure) and whole-song samples (32 measures). For more evaluations, we refer the readers to the appendices.

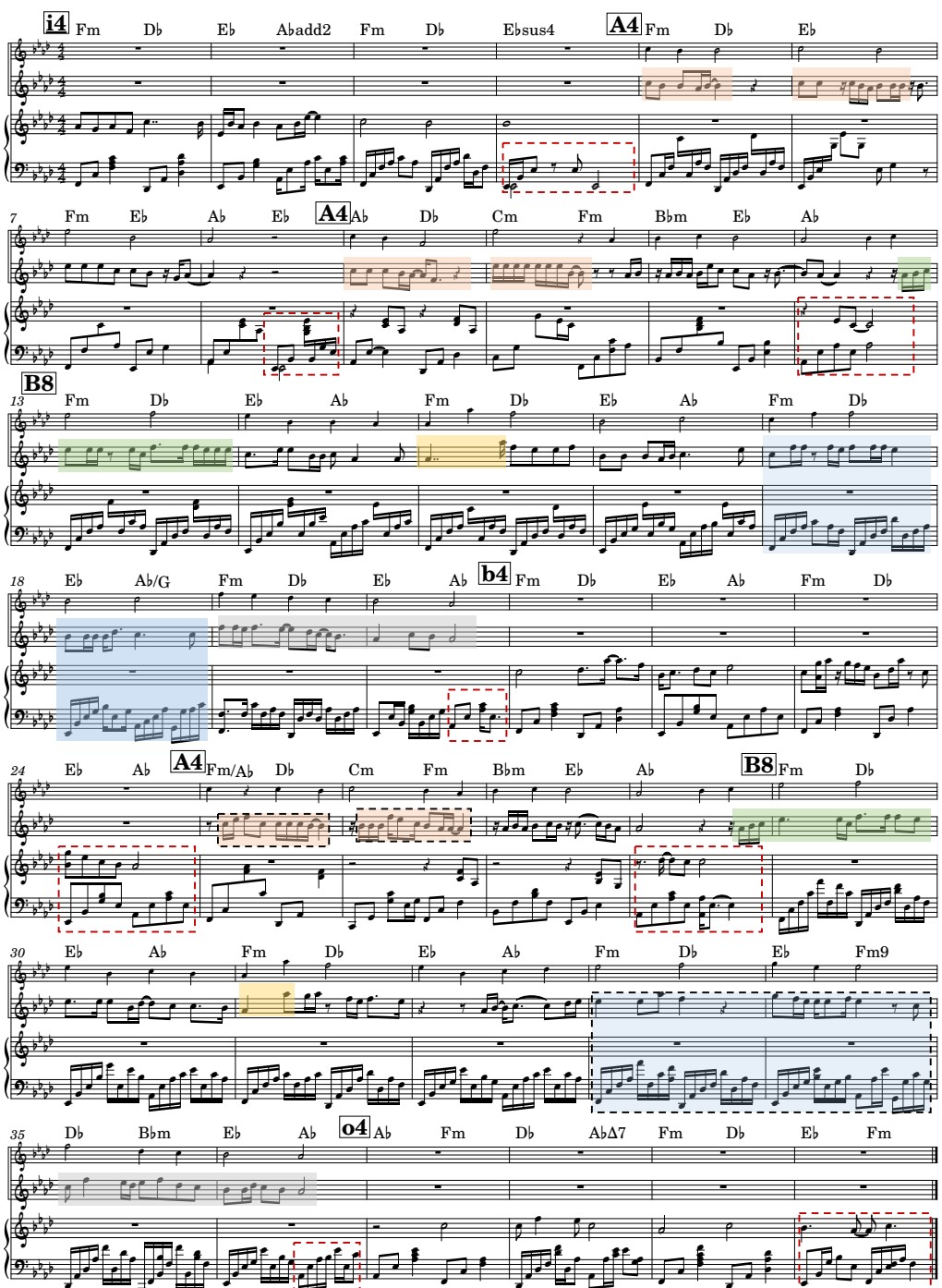

Figure 2: An example of whole-song generation of 40 measures under a given Form (A♭ major key and `"i4A4A4B8b4A4B8o4"` phrases). The three staves (from top to bottom) show the generated Reduced Lead Sheet, Lead Sheet, and Accompaniment. Here, rectangles with colored background are used to indicate the appearance of the same motifs in verse and chorus sections. Dashed border rectangles with colored background indicate a variation of motifs. We use red dotted rectangles to show where the generated score show a strong implication of phrase boundary or cadence. The generated chord progressions in Reduced Lead Sheet and Lead Sheet are identical, shown by the chord symbols.

## 5.1 DATASET

We use the POP909 dataset to train our model (Wang et al., 2020a). POP909 is a pop song dataset of 909 MIDI pieces containing lead melodies, secondary melodies, piano accompaniment tracks, key signatures, and chord annotations. We pad each song to 256 measures to train Stage 1, and segment each song into corresponding time scopes (128 beats for stage 2 and 128 steps for Stage 3 and 4) with a hop size of one measure. 90% of the songs are used for training and the rest 10% are used for testing. Training samples are transposed to all 12 keys. In Appendix A, we introduce more details on data processing.

## 5.2 BASELINE SETTINGS

We construct two baseline models for whole-song Lead Sheet and Accompaniment generation tasks. The two models are modified from two state-of-the-art phrase-level generation models: a diffusion-based one, and a Transformer-based one.

**Diffusion-based** (*Polyff.+ph.l.*). We add phrase label control to Polyffusion (Min et al., 2023) as the external condition, and use the iterative inpainting technique to generate full pieces. We train two separate diffusion models for Lead Sheet and Accompaniment generation on POP909, both adopting the same data representations as the proposed method. This also serves as an ablation study on the effectiveness of our cascaded model design.

**Transformer-based** (*TFxl(REMI)+ph.l.*). Naruse et al. (2022) enables phrase label control on the REMI representation (Huang & Yang, 2020) with Transformer-XL as the model backbone. Similarly, we train two versions for Lead Sheet and Accompaniment generation on POP909.

## 5.3 EVALUATION

**Objective evaluation.** For whole-song well-structuredness, we design the *Inter-Phrase Latent Similarity* (ILS) metric to measure music structure based on content similarity. The metric encourages similarity between music with same phrase labels (e.g., two verses), and distinctiveness between music with different phrase labels (e.g., verse and chorus). We leverage pre-trained disentangled VAEs that encode music notes into latent representations and compare cosine similarities in the latent space. Given a similarity matrix showing pairwise similarity of 2-measure segments within a song, ILS is defined as the ratio between average similarity between phrases of the same types and average similarity between all phrases, and therefore higher values indicate better structure.

Table 2: Objective evaluation of music structure via the proposed inter-phrase latent similarity.

| | Lead Melody | | Chord | Accompaniment |
|---|---|---|---|---|
| | $\text{ILS}^{\text{p}} \uparrow$ | $\text{ILS}^{\text{r}} \uparrow$ | $\text{ILS}^{\text{chd}} \uparrow$ | $\text{ILS}^{\text{txt}} \uparrow$ |
| Ground Truth | $2.28 \pm 0.14$ | $2.30 \pm 0.13$ | $1.42 \pm 0.07$ | $1.68 \pm 0.09$ |
| Cas.Diff. (ours) | $\mathbf{2.05} \pm 0.14$ | $1.49 \pm 0.07$ | $\mathbf{1.32} \pm 0.05$ | $\mathbf{1.19} \pm 0.06$ |
| Polyff. + ph.l. | $0.60 \pm 0.12$ | $0.76 \pm 0.05$ | $0.52 \pm 0.06$ | $0.61 \pm 0.04$ |
| TFxl(REMI) + ph.l. | $1.89 \pm 0.15$ | $\mathbf{1.71} \pm 0.13$ | $0.68 \pm 0.06$ | $0.74 \pm 0.04$ |

We compute ILS on lead melody, chord, and accompaniment. Using pre-trained VAEs from Yang et al. (2019) and Wang et al. (2020b), we compute the latent representations of pitch contour and rhythm (i.e., $z_{\text{p}}, z_{\text{r}}$) for lead melody, latent $z_{\text{chd}}$ for chord, and latent texture $z_{\text{txt}}$ for accompaniment. We pre-define four types of common phrases and let models generate 32 samples for each phrase type, resulting in 128 full songs in total. $\text{ILS}^{\theta}, \theta \in \{\text{p}, \text{r}, \text{chd}, \text{txt}\}$ are calculated for each song, and we show their mean and standard deviation in Table 2. The results show our model significantly outperforms baselines on the phrase content similarity of chord and accompaniment, indicating its effectiveness in preserving long-term structure.

**Subjective evaluation.** We design a double-blind online survey that consists of two parts: short-term (8 measures) evaluation of music quality, and whole-song (32 measures) evaluation of both music quality and well-structuredness. Participants rate *Creativity*, *Naturalness*, and *Musicality* for short-term music segments. For whole-song evaluation, we drop *Creativity* but introduce two more criteria: *Boundary Clarity* and *Phrase Similarity* to focus on the structure of the generation. All

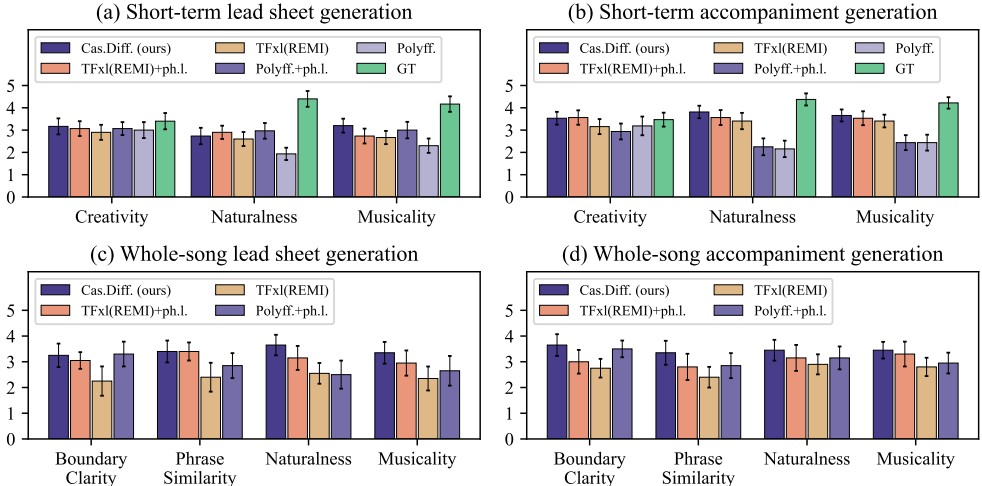

Figure 3: Subjective evaluation results on music quality and well-structuredness. GT indicates ground truth samples composed by humans.

metrics are rated based on a 5-point scale. We constrain whole-song length to 32 measures so that the participants can better memorize the samples and the survey can have a reasonable duration. These generated pieces still preserve a condensed pop-song structure by specifying Form to contain intro, outro and repetitive verses or choruses.

Additionally, for short-term evaluation, we use two more reference models: *Polyff.* and *TFxl(REMI)*, two baseline models without phrase label conditions. This is to investigate whether the introduction of phrase control causes degradation in music quality. For each model, we select three samples for both short-term and whole-song evaluation as well as both lead sheet and accompaniment generation, resulting in $3 \times 2 \times 2 = 12$ groups of samples. Each group of samples shares the same prompt (2 measures for 8-measure samples, and 4 measures for 32-measure samples) and phrase labels (for whole-song evaluation). In the survey, both the group order and the sample order are randomized.

A total of 57 people participated in our survey, and the evaluation result is shown in Figure 3. The bar height shows the mean rating, and the error bar shows the 95% confidence interval (computed by within-subject ANOVA). We show that our model significantly outperforms baselines in the structural metrics of whole-song generation, especially in accompaniment generation. Our model consistently outperforms baselines in terms of music quality in both short-term and whole-song generation, proving that the introduction of compositional hierarchy does not hinder the generation quality.

## 6 CONCLUSION

In conclusion, we contribute the first hierarchical whole-song deep generative algorithm for symbolic music. The current study focuses on the pop music genre, and experimental results demonstrate that our model consistently generates more structured, natural, and musical outputs compared to baseline methods, both at the whole-song and the phrase scales. Additionally, our model offers extensibility, allowing flexible external controls via pre-trained music embeddings. Our approach relies on two key components: a hierarchical music language that balances human interpretability with computational tractability, and a cascaded diffusion architecture that effectively captures the hierarchical structure of entire compositions through both top-down and context-dependent mechanisms. It demonstrates that a strong structural inductive bias can lead to more effective and efficient learning for deep music generative models, and such methodology is potentially useful for other domains as well. In the future, we plan to extend our hierarchical language and generation approach to both multi-track symbolic music and music audio.

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

APPENDICES

The appendices included in this paper are designed to provide more technical details, generation samples, and discussions about our study. They are organized into three parts. Appendices A-B introduce more detailed information about data data representation and processing, model architecture, and training; Appendix C provides some more generation samples; and Appendices D-G include more discussions on the external control efficacy, model's memorization v.s. generation ability, model efficiency, and limitations.

## A    DATA REPRESENTATION AND PROCESSING DETAILS

In this section, we introduce the methods to extract the four levels of music languages defined in Section 3.1. We begin by quantizing MIDI files from the POP909 dataset using the provided beat annotations. For the Form language, we simplify the phrase labeling from Dai et al. (2020) to include only six phrase types (see Table 3 for reference), and extract keys by pitch profile matching, similar to the method described in Krumhansl (2001). In the Reduced Lead Sheet language, melody reduction is computed through a shortest path reduction algorithm,[2] and chords are downsampled by merging consecutive chords that share the same triad structure (root, third, and fifth) within one measure. The Lead Sheet language uses the vocal melody (i.e., "MELODY" track) and the provided chord annotations. The Accompaniment language combines the secondary melody (i.e., "BRIDGE" track) with the accompaniment (i.e., "PIANO" track).

As discussed in Section 3.2, the processed data is converted to image-like data representation. A visualization of the data representation is provided in Figure 4.

Table 3: Definition of phrase type

| Phrase Type | Channel ID | Meaning |
|:---:|:---:|:---:|
| "A" | 0 | Verse section phrases |
| "B" | 1 | Chorus section phrases |
| "X" | 2 | Other phrases with lead melody |
| "i" | 3 | Intro section phrases |
| "o" | 4 | Outro section phrases |
| "b" | 5 | Bridge section phrases |

## B    MODEL ARCHITECTURE AND TRAINING DETAILS

In this section, we elaborate on the model architecture and training. We first discuss the three conditioning methods introduced in Section 3.3 in more detail. The configurations of the four generation stages are summarized in Table 4.

**Detail on background condition.** At each level $k > 1$, the background condition $\boldsymbol{X}_{t:t+b_k}^{<k}$ is represented by an image having the same width and height as the diffusion output. Thus, the background condition can be concatenated with the input along channel axis at each step of the diffusion process. The background condition will be set to all $-1.0$ under the probability $p_{\text{uncond}} = 0.2$, following classifier-free guidance (Baykal et al., 2023).

**Detail on autoregressive condition.** At each level $k > 1$, the generation of $\boldsymbol{X}_{t:t+b_k}^{k}$ is dependent on past generation $\boldsymbol{X}_{<t}^{\leq k}$. Here we select top-$S_k$ past segments of length $b_k'$ based on their *phrase type similarity* to the current segment. These music segments are embedded using a 3-layer 2d-convolutional network and fed to the backbone diffusion models by cross-attention mechanism. In our implementation, we set $S_2 = 3$, $b_2' = 32$, $S_3 = S_4 = 2$, and $b_3' = b_4' = 64$. The autoregressive condition will be set to all $-1.0$ under the probability $p_{\text{uncond}} = 0.1$.

**Detail on external condition.** The condition for Reduced Lead Sheet is four 8-measure latent codes of chord progression encoded from the chord encoder in Min et al. (2023). The condition for Lead

---

[2]Code for melody reduction is available at https://github.com/ZZWaang/melody-reduction-algo.

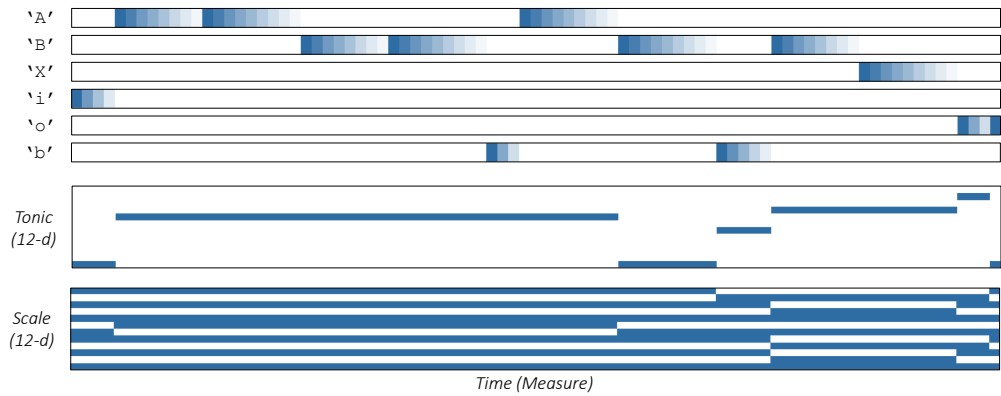

(a) The first-level language Form. The six channels of phrases are shown at the top and brightness indicates phrase countdown (see Equation 1). The two channels of keys are shown at the bottom and brightness indicates one-hot tonic and multi-hot scale.

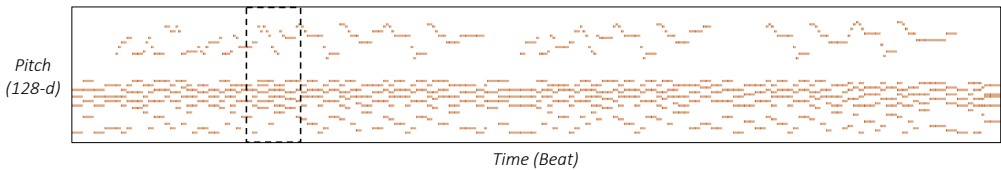

(b) The second-level language Reduced Lead Sheet. Brightness indicates onset and sustain channels.

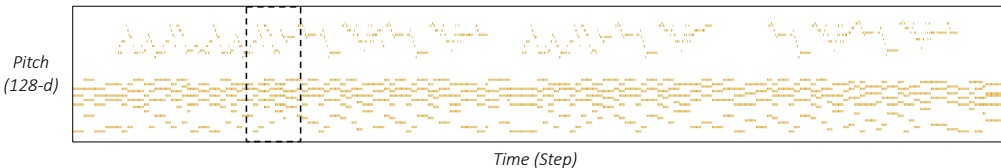

(c) The third-level language Lead Sheet. Brightness indicates onset and sustain channels.

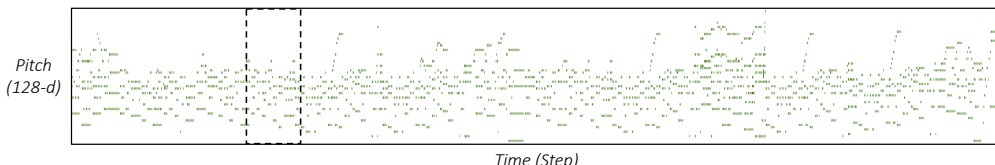

(d) The fourth-level language Accompaniment. Brightness indicates onset and sustain channels.

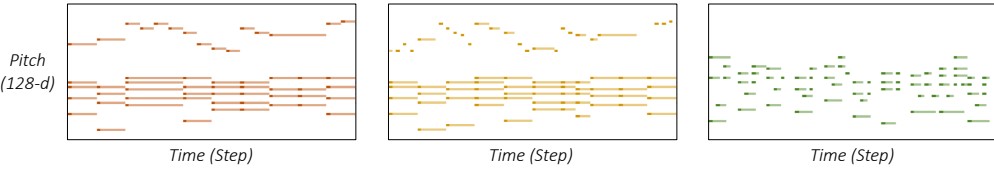

(e) Zoom-in views of the segments in Figure 4b (left), Figure 4c (middle), and Figure 4d (right) marked with rectangles.

Figure 4: An example data representation of our proposed hierarchical music language.

Sheet is four 2-measure latent codes of rhythmic pattern encoded from the $EC^2$-VAE encoder in Yang et al. (2019). The condition for Accompaniment is four 2-measure latent codes of accompaniment texture encoded from the texture encoder in Wang et al. (2020b). These latent codes are fed to the backbone diffusion models by cross-attention mechanism and set to all $-1.0$ under the probability $p_{\text{uncond}} = 0.2$.

The diffusion models for all four stages use the same noise schedule and training methods. Similar to Min et al. (2023), the backbone model is a 2D-UNet model, the encoder and decoder of which contain 4 layers of 2d-convolution with spatial attention at the third and fourth layers. We summarize these common details in Table 5.

Table 4: Configuration of the conditioning methods in four stages of the proposed model.

|  | Form | Red. Lead Sheet | Lead Sheet | Accompaniment |
|---|---|---|---|---|
| Time scope | 256 measures | 128 beats | 128 steps | 128 steps |
| Output shape | $(6, 256, 12)$ | $(2, 128, 128)$ | $(2, 128, 128)$ | $(2, 128, 128)$ |
| Background cond.: shape | N/A | $(6, 128, 128)$ | $(8, 128, 128)$ | $(10, 128, 128)$ |
| Background cond.: $p_{\text{uncond}}$ | N/A | 0.2 | 0.2 | 0.2 |
| Autoreg. cond.: # of seg. | N/A | 3 | 2 | 2 |
| Autoreg. cond.: shape | N/A | $(8, 32, 128)$ | $(10, 64, 128)$ | $(12, 64, 128)$ |
| Autoreg. cond.: $p_{\text{uncond}}$ | N/A | 0.1 | 0.1 | 0.1 |
| Ext. cond.: # of latent codes | N/A | 4 | 4 | 4 |
| Ext. cond.: latent dimension | N/A | 512 | 128 | 256 |
| Ext. cond.: $p_{\text{uncond}}$ | N/A | 0.2 | 0.2 | 0.2 |

Table 5: The hyperparameter configuration of diffusion model training. The listed attributes are the same across all four stages.

| Hyperparameter | Configuration |
|---|---|
| Diffusion Steps ($N$) | 1000 |
| Noise Schedule | Linear from 1 to $1e-4$ |
| UNet Channels | 64 |
| UNet Channel Multipliers | $1, 2, 4, 4$ |
| Batch Size | 16 |
| Attention Levels | $3, 4$ |
| Number of Heads | 4 |
| Learning Rate | $5e-5$ |

## C  MORE EXAMPLES ON STRUCTURAL GENERATION

In this section, we break down each level of the hierarchical language and show more generation examples. For each level, we fix the upper level, and demonstrate a variety of generation results under the upper level control. We also show generation samples that are controlled by external conditions.

**Form generation.** Below shows examples of Form generated by our model:

```
(i8)(A8B16A8B16)(b6)(B14)(o2)
(i12)(A4A4B12)(b4b4)(A4A4B12)(b4b4(B16)o4o1)
(i4)(A4A4B4X5)(b4)(A4b5B4X4X5)(o2)
(i4)(A8B9A8B9X18)(o2o1)
(i8)(A4A4B4B4)(b7)(A4A4B4B4B4B4)(o4)
(i8)(A4A4B4B4)(b7)(A4A4B4B4B4B4)(o4)
(i8)(A8B8X8X8)(b4b4)(A8B8X8X8X4)(o6o1)
(i4)(A4A4B9)(b3)(A4B10B9X1)(o2)
(i4)(A4A4B9)(b3)(A4B10B9X1)(o2)
(i12)(A16B16)(b4b4b4)(A16B16B16)(o10o1)
```

Here, we use parentheses to manually group music sections for better readability. The results show the model captures verse-chorus form of pop songs: the composition usually starts with intro and ends with outro; verse and chorus appear multiple times with bridge phrases in between. Phrases are usually 4 or 8 measures long, similar to real music samples.

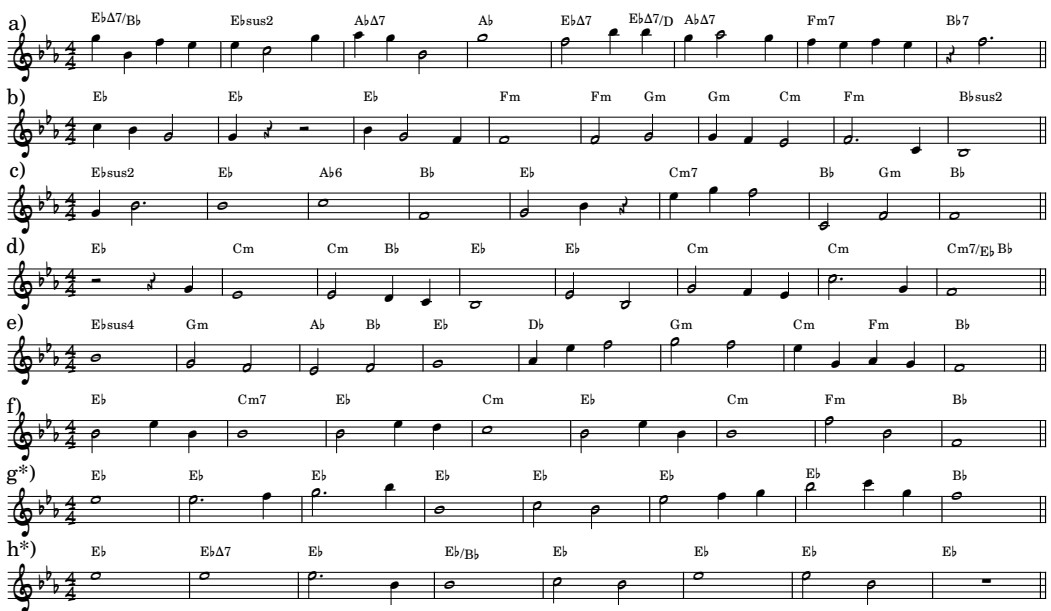

Figure 5: Examples of generated Reduced Lead Sheet of `"A8"` phrase in E♭ major. The samples marked with * are controlled by the external condition meaning "unchanging chord progression".

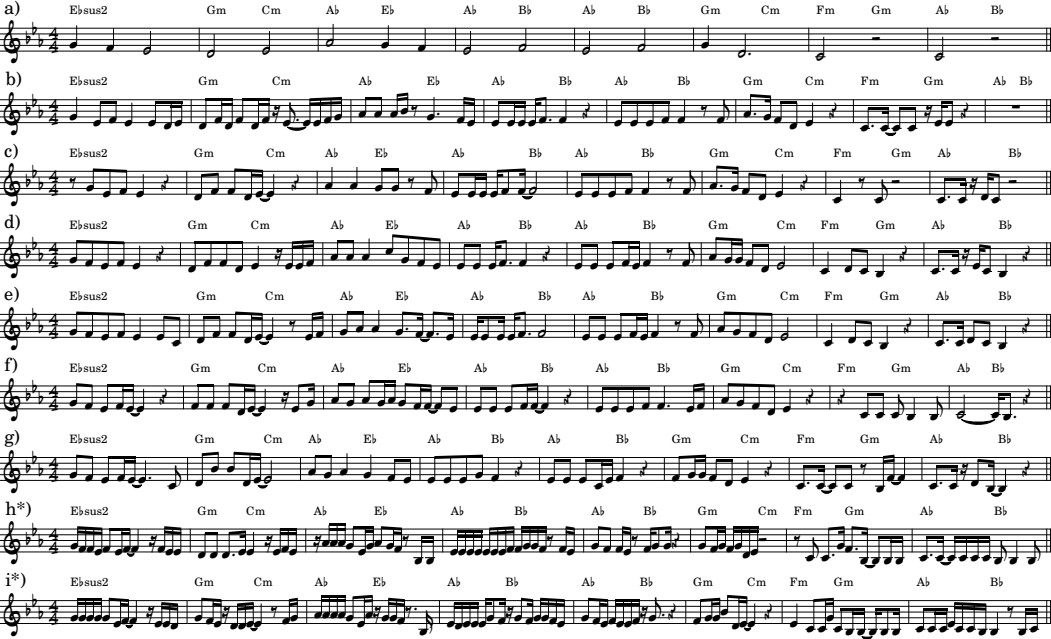

Figure 6: Examples of generated Lead Sheet of `"A8"` phrase in E♭ major given the upper-level Reduced Lead Sheet Figure 6a. The samples marked with * are controlled by the external condition meaning "sixteenth-note rhythm".

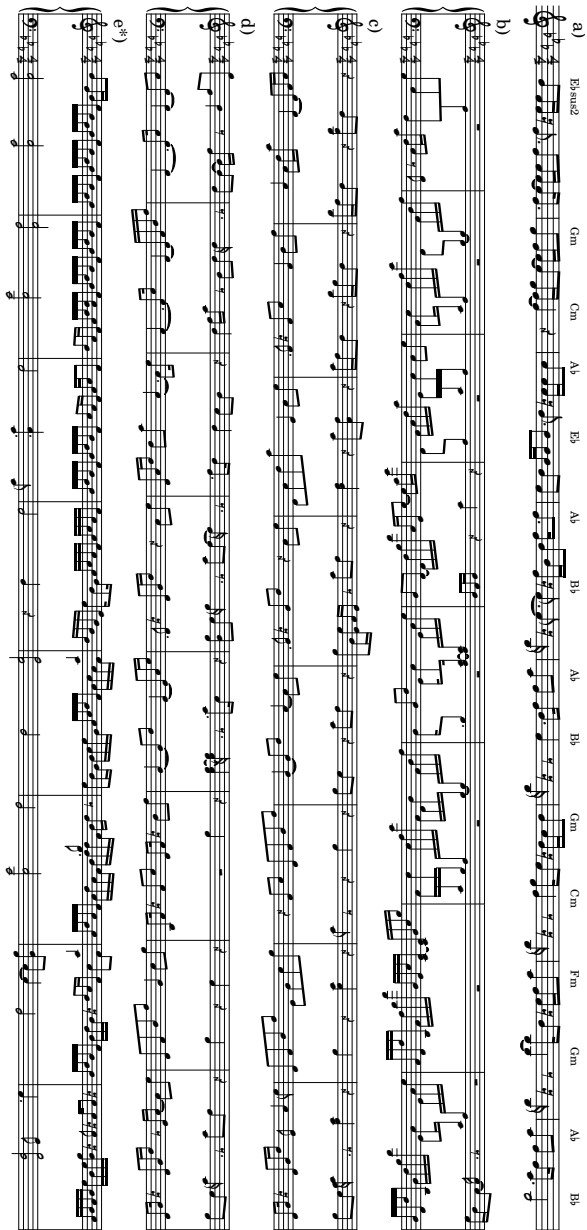

Figure 7: Examples of generated Accompaniment of `"A8"` phrase in E♭ major given the upper-level Lead Sheet Figure 7a. The sample marked with * are controlled by the external condition meaning "Alberti bass".

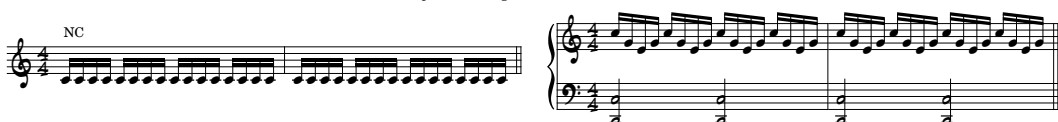

(a) The external music sample used to control the generation of Figure 6h-i.

(b) The external music sample used to control the generation of Figure 7e.

Figure 8: The music samples for external control.

**Reduced Lead Sheet generation with external harmony control.** Figure 5a-f show examples of 8-measure generation of the Reduced Lead Sheet level. The results are all controlled by the same Form: an 8-measure verse phrase in E♭ major key. The generated samples show many ways to develop the melody (different contour and melodic climax positions) and the harmony (different chord types and harmonic rhythm). Moreover, each of the samples has a consistent style and usually ends in a tonic or dominant chord indicating the ending of a phrase. Moreover, we also apply the external control of "unchanging chord" to generate Figure 5g-h. Such a control is achieved by encoding the latent chord representation of a sequence of all `Eb:maj` chords using the pre-trained VAE encoder (Min et al., 2023). The results have fewer changes in harmony and the melody reduction alters accordingly.

**Lead Sheet generation with external rhythm control.** Figure 6b-g show examples of 8-measure Lead Sheet generation controlled by the same Reduced Lead Sheet, shown in Figure 6a. The generated samples follow the pitch contour in the melody reduction and differ in local pitch and rhythm patterns. At this level, we also use latent control of "sixteenth-note rhythm" to generate Figure 6h-i. Such a control is achieved by encoding the latent rhythm representation of the sample in Figure 8a using the pre-trained VAE encoder (Yang et al., 2019). The generation examples show melody realization with more frequent onsets accordingly.

**Accompaniment generation with external texture control.** Figure 7b-d show examples of 8-measure Accompaniment generation controlled by the same Lead Sheet, shown in Figure 7a. The generated samples mainly use arpeggios but are different in the exact patterns. Some of the generation has a "fill" in the fourth and eighth measures to indicate phrasing. At this level, we also use latent control of "Alberti bass" to generate Figure 7e. Such a control is achieved by encoding the latent texture representation of the sample in Figure 8b using the pre-trained VAE encoder (Wang et al., 2020b). The generation adopts the Alberti bass figure and makes reasonable variations.

## D   EVALUATION OF EXTERNAL CONTROL EFFICACY

In this section, we evaluate the efficacy of external controls. These controls are achieved by feeding pre-trained representations as external condition to each layer of the cascaded diffusion model (introduced in Section 3.3). Specifically, we evaluate three scenarios: (1) chord control in Reduced Lead Sheet generation (Stage two), (2) rhythm control in Lead Sheet generation (Stage three), and (3) texture control in Accompaniment generation (Stage four). In this section, we let $z^{\text{ext}}$ denote the external control in one of the three scenarios and let $x^{\text{ext}}$ denote the actual observation from which $z^{\text{ext}}$ is encoded from. Let $x^{\text{out}}$ denote the conditional generation results.

Efficacy of control can be evaluated by computing the similarity between the input control and the generation result. We propose a *rule-based metric* and a *latent metric*. The rule-based metric directly computes the distance between $x^{\text{out}}$ and $x^{\text{ext}}$ in terms of the corresponding features. In particular, for chord control, we compute the $\ell_2$ distance between the given chord condition and the generated chord at each time step; and for rhythm or texture control, we compute the $\ell_2$ distance of note onsets between the given control and the generated lead sheets or accompaniments. Such distance-based metric has previously been used to evaluate control efficacy in Ren et al. (2020) and Min et al. (2023). In the latent metric, we encode the generation $x^{\text{out}}$ back to the latent code $z^{\text{out}}$ using the same pre-trained encoders and measure the cosine similarity between $z^{\text{out}}$ and $z^{\text{ext}}$.

There are two reference methods for comparison. First, we use the unconditional mode of our method to serve as a baseline where control is ineffective. Second, we generate samples by sampling from the Variational Autoencoders (VAEs) that the pre-trained encoders belong to. Specifically, we sample $z^{\text{ext}}$ from the VAE posterior distribution and sample the rest of the latent codes from Gaussian prior to decode results. By the well-disentangled property shown in the original paper (Yang et al., 2019; Wang et al., 2020b), this reference method indicates the maximum attainable level of controllability.

For each of the three scenario, we randomly selected 32 versions of external control from the test set and generate 128 music segments for each methods. In Table 6, we show the rule-based distance (denoted by $\text{dis}^{\text{rb}}$) and latent similarity (denoted by $\text{sim}^{\text{lt}}$) for the three generation stages. Experimental results show that the use of external condition significantly yields controllability for all three scenarios.

Table 6: Objective evaluation of external control efficacy of chord, rhythm and texture in the three diffusion stages. $\mathrm{dis}^{\mathrm{rb}}$ denotes the rule-based distance-based metric and $\mathrm{sim}^{\mathrm{lt}}$ denotes the latent similarity-based metric.

| | Stage 1: Chord | | Stage 2: Rhythm | | Stage 3: Texture | |
|---|---|---|---|---|---|---|
| | $\mathrm{dis}^{\mathrm{rb}} \downarrow$ | $\mathrm{sim}^{\mathrm{lt}} \uparrow$ | $\mathrm{dis}^{\mathrm{rb}} \downarrow$ | $\mathrm{sim}^{\mathrm{lt}} \uparrow$ | $\mathrm{dis}^{\mathrm{rb}} \downarrow$ | $\mathrm{sim}^{\mathrm{lt}} \uparrow$ |
| Cas.Diff. (uncond) | $2.09 \pm 0.80$ | $0.37 \pm 0.09$ | $2.27 \pm 0.53$ | $0.14 \pm 0.23$ | $3.94 \pm 1.46$ | $0.02 \pm 0.11$ |
| VAE Sampling | $0.19 \pm 0.47$ | $0.97 \pm 0.07$ | $0.14 \pm 0.42$ | $0.96 \pm 0.04$ | $0.33 \pm 0.59$ | $0.90 \pm 0.06$ |
| Cas.Diff. (cond) | $1.73 \pm 1.02$ | $0.48 \pm 0.14$ | $1.10 \pm 0.74$ | $0.75 \pm 0.16$ | $0.87 \pm 0.80$ | $0.89 \pm 0.06$ |

## E   DOES THE MODEL JUST COPY THE TRAINING DATA?

In generative modeling, a critical consideration is whether the model overfits and the generation copies the training data. This section includes a quantitative evaluation focused on the similarity between generated segments and the entire training set. We primarily focus on *melody* similarity, a most recognizable aspect of music composition.

Our goal is to measure the *Degree of Copying (DoC)* with respect to a set of generated samples. Let $x$ be a two-measure melody segment from a generated piece, we define *Similarity to the Training Set* of segment $x$ as:

$$S(x) := \max_{x' \in \mathcal{T}} \mathrm{sim}(x, x'), \tag{7}$$

where $\mathcal{T}$ denotes the training set, $x'$ is a two-measure segment from the training set, and $\mathrm{sim}(x, x')$ computes the similarity between $x$ and $x'$. Here $S(x) \in [0, 1]$, and a larger $S(x)$ shows a higher degree of copying. The DoC can be represented by the histogram of $S(x)$. In our experiments, we report the mean and standard deviation of the histogram. We consider a rule-based similarity metric and a latent similarity metric as follows:

**Rule-based similarity metric.** We compute the note-wise similarity between two segments by matching the exact onsets and pitches. Let $n_{x \cap x'}$ denote the number of notes that appear in both $x$ and $x'$ with the same *pitch class* and *onset*, and let $n_x$ and $n_{x'}$ denote the number of notes in $x$ and $x'$, respectively. The rule-based similarity metric is defined as:

$$\mathrm{sim}^{\mathrm{rb}}(x, x') := \frac{2n_{x \cap x'}}{n_x + n_{x'}}. \tag{8}$$

**Latent similarity metric.** We also measure the melodic similarity in the latent space because rule-based methods cannot detect *indirect copying* (e.g., same pitch contour or rhythm). We leverage the pre-trained $\mathrm{EC}^2$-VAE (Yang et al., 2019), which learns a semantically meaningful and disentangled latent space of pitch contour and rhythmic pattern. We extract the latent code of pitch (denoted as $z_{\mathrm{p}}^x$) and rhythm (denoted as $z_{\mathrm{r}}^x$) of melody segments and compute the cosine similarity in terms of both pitch and rhythm:

$$\mathrm{sim}_{\mathrm{p}}^{\mathrm{lt}}(x, x') := \frac{\langle z_{\mathrm{p}}^x, z_{\mathrm{p}}^{x'} \rangle}{||z_{\mathrm{p}}^x|| \cdot ||z_{\mathrm{p}}^{x'}||}, \tag{9}$$

$$\mathrm{sim}_{\mathrm{r}}^{\mathrm{lt}}(x, x') := \frac{\langle z_{\mathrm{r}}^x, z_{\mathrm{r}}^{x'} \rangle}{||z_{\mathrm{r}}^x|| \cdot ||z_{\mathrm{r}}^{x'}||}. \tag{10}$$

For both of the metrics, the samples in the training set are transposed to 12 keys to account for relative pitch similarity. Segments that only contain rests are discarded beforehand.

In Table 7, we show the mean and standard deviation of $S(x)$ on the data samples generated using our proposed methods and other baselines used for whole-song generation. We compute the statistics of the test set of POP909 as a reference for *no risk of copying*, since no song in the training set (or their cover-song versions) appears in the test set. We also design two copy-bots as references for *potential risk of copying*. The first copy-bot copies a different part of the training set at each measure, which emulates a *direct copying* behavior. The second copy-bot encodes the melodies from the

Table 7: Evaluation on whether the generative models copy the training data. The highlighted data in red indicate potential copying the training set.

| Sample Source | Sample Size | Similarity Metric | | |
|---|---|---|---|---|
| | | $\text{sim}^{\text{rb}} \downarrow$ | $\text{sim}^{\text{lt}}_{\text{p}} \downarrow$ | $\text{sim}^{\text{lt}}_{\text{r}} \downarrow$ |
| Test set *(no plag.)* | 88 pieces | $0.6567 \pm 0.1141$ | $0.8637 \pm 0.0486$ | $0.8320 \pm 0.0680$ |
| Copy-bot 1 *(plag.)* | 128 pieces | $0.7108 \pm 0.1159$ | $0.8616 \pm 0.0526$ | $0.8276 \pm 0.0699$ |
| Copy-bot 2 *(plag.)* | 128 pieces | $0.6888 \pm 0.1628$ | $0.9086 \pm 0.0340$ | $0.8555 \pm 0.0411$ |
| Cas.Diff. (ours) | 128 pieces | $0.6530 \pm 0.1321$ | $0.8743 \pm 0.0491$ | $0.8180 \pm 0.0710$ |
| Polyff. + ph.l. | 128 pieces | $0.6117 \pm 0.1162$ | $0.8639 \pm 0.0487$ | $0.8424 \pm 0.0622$ |
| TFxl(REMI) + ph.l. | 128 pieces | $0.6088 \pm 0.1053$ | $0.8599 \pm 0.0446$ | $0.8154 \pm 0.0642$ |

*training set and adds noise to the latent representation before reconstruction, which emulates an indirect copying behavior. Experimental results show that our proposed method (as well as the baseline whole-song generation methods) have similar DoC compared to that of the test set. Also, the proposed metrics successfully detect both direct and indirect copying behaviors as the DoCs of copy-bots are noticeably higher. Thus, we conclude that our model has a very low risk of copying the training set.*

## F DISCUSSION: EFFICIENCY OF THE CASCADED METHOD

In this section, we conduct a theoretical comparison between the efficiency of our cascaded diffusion models and an alternative end-to-end approach, which generates a full-piece music with a single diffusion model. We evaluate the *time* and *model parameter* complexities of both methods. As summarized in Table 8, we demonstrate that end-to-end approaches exhibit quadratic complexities in both time and model parameters relative to data length, whereas the cascaded approach achieves linear time complexity and logarithmic model parameter complexity.

We base our analysis on a diffusion architecture with a UNet backbone, identical to the proposed architecture. We assume the sequential data has length $T$ (corresponding to image width in UNet), and the dimension of each time step is $D$ (corresponding to image height in UNet). The UNet will first embed the input image to have shape $(C, T, D)$ regardless of the number of input channels, where $C$ is called the base channel size. Additionally, our computation assumes a typical UNet configuration as follows:

1. In the contracting path of the UNet, the number of channels at each layer doubles, and the width of the feature maps are halved due to max pooling.

2. The expanding path of the UNet mirrors the contracting path.

3. The number of the levels (or model depth) $M_{T'}$ scales logarithmically with the input sequence length $T'$, i.e., $M_{T'} = \log_2 T' + c_0$, where $c_0$ is a constant.

**Complexity of end-to-end approach.** At each level $i = 0, \ldots, M_T = \log_2 T + c_0$, the width of the feature map is $\frac{T}{2^i}$, the number of input channels is $C \cdot 2^i$, and the number of output channels is $C \cdot 2^{i+1}$. The time complexity of the convolution per layer can be computed as:

$$\mathcal{O}\big((\frac{T}{2^i} \cdot D) \cdot (C \cdot 2^i) \cdot (C \cdot 2^{i+1})\big) = \mathcal{O}(T \cdot D \cdot C^2 \cdot 2^{i+1}) = \mathcal{O}(T \cdot 2^{i+1}). \tag{11}$$

In Equation 11, we regard $D$ and $C$ as constants, therefore removing from the complexity term. Summing Equation 11 over all levels results in the total time complexity:

$$\sum_{i=0}^{M_T} \mathcal{O}(T \cdot 2^{i+1}) = \mathcal{O}(T^2). \tag{12}$$

Similarly, at each level $i = 0, \ldots, M_T$, the number of model parameters can be computed as:

$$\mathcal{O}\big((C \cdot 2^i) \cdot (C \cdot 2^{i+1})\big) = \mathcal{O}(C^2 \cdot 2^{2i+1}) = \mathcal{O}(2^{2i+1}), \tag{13}$$

Table 8: Theoretical comparison of time and model parameter complexity between cascaded approach and end-to-end approach. $\eta$ denotes the resolution scaling factor. $L$ denotes the time scope (i.e., receptive field) of cascaded models.

| | Cascaded Approach | End-to-End Approach |
|---|---|---|
| Time Complexity | $\mathcal{O}(L^2 T)$ | $\mathcal{O}(T^2)$ |
| Model Parameter Complexity | $\mathcal{O}(L^2 \log_\eta T)$ | $\mathcal{O}(T^2)$ |

resulting in the total model parameter complexity:

$$\sum_{i=0}^{M_T} \mathcal{O}(2^{2i+1}) = \mathcal{O}(T^2). \tag{14}$$

In conclusion, the time and model parameter complexity for end-to-end approach are both $\mathcal{O}(T^2)$.

**Complexity of cascaded approach.** The cascaded models proposed in this paper is tailored for music data. For the analysis in this section, we define a theoretical cascaded approach as follows. To generate a sequential data of length $T$, we define a $K$-level compositional hierarchy, and the resolution at each level is scaled by a factor $\eta$. The top-level ($k = 1$) has a resolution of $L$, and for each level $k = 2, \ldots, K$, the resolution is $L \cdot \eta^{k-1}$, such that at the final level $K$, the resolution is $L \cdot \eta^{K-1} = T$. We train $K$ diffusion models in total, all having the same generation scope $L$. Thus, the diffusion model at level $k = 1$ generates the whole sequence, and the diffusion models for level $k = 2, \ldots, K$ generate only a slice of the sequence. The model parameter complexity of a single level can be computed by substituting $M_T$ with $M_L$ in Equation 14:

$$\sum_{i=0}^{\log_2 L + c_0} \mathcal{O}(L \cdot 2^{i+1}) = \mathcal{O}(L^2). \tag{15}$$

Summing up the parameters in $K$ separate models results in the total model parameter complexity:

$$\mathcal{O}(L^2 \cdot K) = \mathcal{O}(L^2 \log_\eta T). \tag{16}$$

Similarly, for time complexity, each model call is $\mathcal{O}(L^2)$ (see Equation 12); and at all levels, the generation requires $(2\eta^{k-1} - 1)$ autoregressive iterations (see Algorithm 1). So, the total time complexity is:

$$\sum_{k=1}^{K} \mathcal{O}(L^2 \cdot \eta^{k-1}) = \mathcal{O}(L^2 T). \tag{17}$$

Therefore, the cascaded approach has a time complexity of $\mathcal{O}(L^2 T)$ and a model parameter complexity of $\mathcal{O}(L^2 \log_\eta T)$. The computation implies that the efficiency of cascaded approach will be more significant when $T >> L$. Theoretically, if $L$ is a constant (e.g., bounded by the computational resources available), the time and model parameter complexity will become $\mathcal{O}(T)$ and $\mathcal{O}(\log_\eta T)$, respectively.

# G  LIMITATIONS AND FUTURE PLAN

Our current model supports a maximum generation scope of 256 measures, typically adequate for pop songs, but insufficient for other genres (e.g., classical music), which may require longer lengths. While our model can generate irregular phrase lengths and theoretically supports both 3/4 and 4/4 time signatures, it does not support meter change, and the capability of 3/4 song generation is limited, since the proportion of 3/4 songs in the dataset is pretty low. Additionally, we observe that the generated endings are sometimes not ideal, such as failing to resolve on the tonic harmony. We suspect the issue is related to imprecise quantization in the POP909 dataset's outro sections. Moreover, there is room for improvement in overall music quality, as the model occasionally produces obvious flawed samples, such as blank measures. To address these issues, we plan to enhance model performance with a more refined architecture and more extensive data training.

