# OpenReview forum: "Whole-Song Hierarchical Generation of Symbolic Music Using Cascaded Diffusion Models"
_ICLR.cc/2024/Conference — ICLR 2024 spotlight_

### Official Review · Reviewer_EK1V · 2023-10-31

**Soundness:** 3 good
**Presentation:** 4 excellent
**Contribution:** 3 good
**Rating:** 8
**Confidence:** 5

**Summary:**

This paper proposes a hierarchical strategy for symbolic music generation. Specifically, their approach involves a cascade of conditional diffusion models which iteratively generate a series of interpretable representations in a coarse-to-fine fashion. The authors compare their proposed approach to strong baselines through both quantitative metrics and a qualitative user study, demonstrating promising performance.

**Strengths:**

This is a nice paper overall. Among its virtues are (1) **the quality of the results**, (2) **simplicity of the approach**, (3) **usefulness for controllable generation**, and (4) **clarity of the writing**.

**Result quality**. The proposed method achieves impressive results in both the quantitative evaluation and subjective tests, especially compared to strong baselines. Moreover, the included sound examples are quite compelling, and the contribution of the proposed hierarchical approach to the final outputs is immediately apparent.

**Simplicity**. While the design of the hierarchical approach is somewhat complex, the proposed generative modeling approach is satisfyingly simple, with each stage using the same basic setup despite their structural differences, and later stages adding in well-motivated mechanisms to address clear issues (e.g., autoregressive component to generate locally but with global consistency).

**Usefulness for control**. The proposed hierarchy (both in the data representation and modeling) is helpful for enabling long-form generation w/ global structure. However, it has an additional benefit of enabling interpretable manipulation of intermediate representations. While the authors don’t specifically explore interaction, it is clear that this aspect of the approach could be very powerful for users.

**Well-written**. This paper is extremely clear and well-written, especially relative to the median paper on music generation. Symbolic music generation, especially work that focuses on interpretability, tends to be a very messy subject with lots of in-the-weeds details that often manifest as confusing and poorly-written papers. All symbolic music gen papers tend to require substantial music expertise to fully understand, but this paper does a fantastic job of both minimizing the expertise needed and being exceptionally clear in overall formulation.

**Weaknesses:**

There are two primary weaknesses with this work: (1) **unclear if model is copying**, and (2) **impact is limited by data availability**

**Unclear if model is copying**. This model is trained on a very small amount of data, just 909 songs. Despite this, subjectively speaking, the results from the proposed model are quite good. It seems quite plausible that the model is overfit to the training data and producing copies or near-copies. Can the authors provide any evidence that this is not happening?

**Impact limited by data availability**. A broader issue is that extracting the proposed hierarchical representation requires rich annotations aligned with the raw notes: chords, melody, key, phrase boundaries, etc. Some of this the authors extract (e.g. key / phrase boundaries) and some of this comes from human labels (e.g. chords and melody). This limits the overall applicability of the approach to music datasets with such rich labels (POP909), and prevents application to much larger symbolic music datasets (Lakh) which cover more styles.

Low-level comments:

- Table 1 could describe M, γ, δ in the caption
- Table 1 would be well-complemented by a Figure 1 showing the languages visually and their relationships to music scores / one another
- Sound examples page could have qualitative examples from baselines
- It’s a shame that there is no “overall structure” music in the whole-song subjective study, so users can rate if models produce music with clear long-form structure.

**Questions:**

Can the authors provide quantitative or qualitative evidence that the model is not overfit to the training data? If so, I would consider raising my score from a 6 to an 8.

---

> ### Author Response · Authors · 2023-11-19
> **Response to Reviewer EK1V**
>
> Thank you for your careful review and acknowledging our contributions. In this rebuttal, we will respond to the weakness and questions: 1) plagiarism and overfitting issue and 2) dataset limitation. We also thank you for leaving some low-level comments which we will consider in paper revision.
>
> ### **Plagiarism issue**
>
> Thank you for pointing out this issue! Since this is really important and is also mentioned by other reviewers, we write a response to all reviewers to answer the question. Please kindly check the **response to all reviewers** for more detail. In short, the current results show that our model does not suffer from plagiarism problem.
>
> ### **Dataset limitation**
>
> Currently, we use POP909 to study whole-song generation because our current focus is on pop songs and the dataset contains relatively accurate annotation. With the help of current MIR algorithms, we can achieve not only phrase structure analysis, but also score quantization, melody extraction and so on. (Actually most of the annotation from the POP909 dataset is extracted using algorithms.) In our ongoing project, we are extracting annotations from larger MIDI datasets and train our model on these datasets. We expect the performance will be better with more training data.
>
> We admit there is a limitation for our current methods that our hierarchical language only works for pop songs (e.g., not for classical) and does not directly work for multi-track music. In our future work, we hope to define more general hierarchical languages to model those types of music.

---

> > ### Comment · Reviewer_EK1V · 2023-11-21
> > **Changed score**
> >
> > Thank you for the response. I am reasonably satisfied by the initial plagiarism analysis and explanation about applicability to larger datasets, and have increased my score. I remain surprised that results can be this strong given the small size of the dataset, and encourage the authors to eventually add a more detailed plagiarism analysis to their paper as promised.

---

> > > ### Author Response · Authors · 2023-11-22
> > > **Response to Reviewer EK1V**
> > >
> > > Dear Reviewer EK1V,
> > >
> > > Thank you for your reply. And if you haven't already, I welcome you to check the revised paper (already updated on the pdf link) for more detailed plagiarism analysis as well as the reply to all reviewers titled "Paper revision uploaded".
> > >
> > > We will make the other changes you mentioned in the future revision.

---

### Official Review · Reviewer_JPv8 · 2023-10-31

**Soundness:** 3 good
**Presentation:** 3 good
**Contribution:** 3 good
**Rating:** 8
**Confidence:** 3

**Summary:**

This paper proposes to learn to generate a full pop song with piano accompaniment as a hierarchical generation process. This paper defines a music language or representation of 4 levels. The generation process is then defined into four stages: form, counterpoint, lead sheet and accompaniment. In each stage, a diffusion model is used as the backbone generative model, generating the image-like representation at that level.

**Strengths:**

1. This paper proposes a novel hierarchical representation of symbolic music.
2. This paper proposes a novel task formulation of generating full-song symbolic music and has a good qualitative result and offers a wide range of controlability.

**Weaknesses:**

The system, including its condition input, appears complex. The reliance on multiple pretrained models (as referenced in Section 3.3) might be cumbersome. It would enhance the paper's credibility if the authors could provide ablation studies both on the model's architecture and the efficacy of the control input.

**Questions:**

I wonder how this method applies to a more general MIDI-like dataset. Or does it rely heavily on the POP909 dataset?

---

> ### Author Response · Authors · 2023-11-19
> **Response to Reviewer JPv8**
>
> Thank you for your review and we are glad to see the strengths of the paper is well-summarized. In this rebuttal, we mainly respond to the weakness and questions.
>
> ### **System complexity**
>
> We admit our system is indeed complex. In this paper, we believe the r*ealization of compositional hierarchy* (the main contribution of our work) is the “skeleton” of music composition; and on top of it we add *autoregressive conditions* and *external conditions* as “plugins”. these “plugins” are necessary and currently they are modeled in a simple way. In the future, we hope *autoregressive condition* can be extended to represent motif development and *external condition* can be developed into a hierarchical cross-modal control interface.
>
> ### **Ablation studies on model’s architecture**
>
> In our experiments, we ablated the assumption whether compositional hierarchy is necessary. We compared our proposed model with two types of baselines:
>
> 1. Models without compositional hierarchy at all. This corresponds to the baselines: Polyff. and TFxl(REMI).
> 2. Models with very simple compositional hierarchy (i.e., only phrase label as high-level language). This corresponds to the baselines: Polyff.+ph.l. and TFxl(REMI)+ph.l.
>
> Both assumptions are implemented with diffusion models (more suitable to the current task) and Transformers (more conventional).
>
> Moreover, we ablate “autoregressive condition” in our preliminary experiments. We trained a version with only compositional hierarchy but no autoregressive condition. The generated result is promising but only within 128 beats (the time scope of stage 2). For example, if two verse sections are far apart (e.g., 40 measures apart), the realization of the Counterpoint will be very different and so the melody lacks coherency. That’s why we incorporate autoregressive condition in our model architecture.
>
> We see that there are other possible ways to define compositional hierarchy and it is important for this study to try more settings to ablate. Please let us know if there are other ablations you suggest and we can try to conduct the experiments as soon as possible.
>
> ### **Ablation studies on the efficacy of the control input**
>
> We train our model under both settings: 1) with external condition (under classifier-free guidance) and 2) without external condition. The performance of both models are similar when doing unconditional generation (without specifying latent codes of external conditions). The fact is also shown in the original classifier-free guidance paper [1].
>
> The effectiveness of the external condition is studied and shown in generation example in section A.4 (Figure 5 (g)(h), Figure 6 (h)(i), and Figure 7(e)). In the Polyffusion paper [2], the authors already conducted systematic experiments to show the control efficacy in diffusion-based music generation models. If we have more time, we can do systematic experiments for whole-song generation.
>
> ### **Generalizability to other MIDI datasets**
>
> Currently, we use POP909 to study whole-song generation because our current focus is on pop songs and the dataset contains relatively accurate annotation. With the help of current MIR algorithms, it is possible to annotate larger MIDI datasets and train our model on these datasets, which is actually our ongoing project. We expect the performance will be better with more training data.
>
> We admit there is a limitation for our current methods that our hierarchical language only works for pop songs (e.g., not for classical) and does not directly work for multi-track music. In our future work, we hope to define more general hierarchical languages to model those types of music.
>
> [1] Ho J, Salimans T. Classifier-free diffusion guidance. arXiv preprint arXiv:2207.12598. 2022 Jul 26.
>
> [2] Lejun Min, Junyan Jiang, Gus Xia, Jingwei Zhao: Polyffusion: A Diffusion Model for Polyphonic Score Generation with Internal and External Controls. ISMIR 2023.

---

> > ### Comment · Reviewer_JPv8 · 2023-11-22
> > **Thanks for the response**
> >
> > Sorry for the late reply. Thank you for the response and explanation. For the efficacy of control: Thanks for pointing out the experiment results in section A.4. What I referred to was conducting experiments on getting quantitative statistics of the qualitative results in the comparisons in section A.4. That could be more convincing in showing the efficacy of the control.

---

> > > ### Author Response · Authors · 2023-11-22
> > > **Response to Reviewer JPv8**
> > >
> > > Dear Reviewer JPv8,
> > >
> > > Thank you for your reply. We will conduct the experiment regarding control efficacy as soon as possible.

---

### Official Review · Reviewer_JZXS · 2023-11-01

**Soundness:** 3 good
**Presentation:** 2 fair
**Contribution:** 2 fair
**Rating:** 5
**Confidence:** 4

**Summary:**

The authors showcase a system to generate complete pop songs in pianoroll format. The approach consists in splitting the generation into a hierarchy of four stages. The representation for each of this stage can be computed directly from the original pianoroll and a bespoke algorithm termed Tonal Reduction Algorithm is introduced to compute the so-called "Conterpoint" representation.
The different level of the hierarchy are then generated iteratively by conditioning on the preceding stages using a diffusion model.

**Strengths:**

This article features a very well-engineered system. The generated pop songs are convincing and seem to capture well the style of the POP909 dataset.
The presentation website features lots of interesting examples that sound well.
It shows that this model is able to cover many use cases beyond generation from-scratch: from accompaniment generation based on texture to leadsheet generation.

**Weaknesses:**

The main weaknesses seem the lack of details concerning the diffusion model and the sampling procedure.
This is even more problematic as it seems that the modeling process is not standard, with the diffusion model used to generate chunks of music in an autoregressive manner.

It is also unclear how the data is represented as it seems at first sight that the diffusion model used would be a Discrete diffusion model.
As such, it is not very accessible for people knowing the standard literature on diffusion models.
"We represent key by K ∈ R2×M×12, where tonic information and scale information are stored on the two channels"
It may be interesting to emphasize on the non standard points .


Very custom Tonal Reduction Algorithm. Seems very close from Polyffusion.
The results shown here may not be of interest for the broad ICLR community as the main contributions are mostly about symbolic music generation in a pianoroll format.

**Questions:**

The diffusion model used auto regressively

What are the pretrained models for chord progression, rhythmic pattern, accompaniment texture?

"Conterpoint" term for the second stage is not appropriate as this stage describes the sketch of the melody and harmony t. "Draft" or "Sketch" stage may be more suitable?

Leadsheet encoded as melody and chords? Are chords in string format?

What is the sampling time for a whole song?

---

> ### Author Response · Authors · 2023-11-19
> **Response to Reviewer JZXS (part one)**
>
> Thank you for your review and insight. In this rebuttal, we address the weaknesses in detail (in part one) and respond to the questions (in part two).
>
> ### **Weakness 1: non-standard diffusion model and unclear data representation**
>
> The diffusion model used in our method is the standard DDPM [1]. Polyffusion [2] has shown such method is an effective way to generate 8-measure music by regarding piano-roll as real-valued image data. (The training process is the same as DDPM. In inference the output of the generation will be quantized to discrete values and empirically such quantization operation does not yield syntax error [2].) In this study, we use the same method to represent music and its high-level languages. Data representation is discussed in section 3.1-3.2 and also in A.2. If there is other details about the model and data representation you think is missing, please let us know. We are happy to explain and make it clearer in the revised version.
>
> ### **Weakness 2: Limited contribution for ICLR community**
>
> Hierarchical music generation is the core problem and we propose to bring innovative methodology to solve it by 1) defining a music compositional hierarchy, and 2) training a corresponding cascaded generative model. The methodology is efficient in data usage and computational resource, and is more controllable and interpretable compared to other existing music generation methods. In this paper, we see Polyffusion as a tool to realize this goal. The Tonal Reduction Algorithm is a small but important design to make sure the hierarchical model is functioning.
>
> For the ICLR community, hierarchical symbolic music generation, as a sequence generation problem, has a unique significance. As stated by Prof. LeCun [3], learning multiple levels of abstraction and hierarchical prediction is one of the main challenges to address for today’s AI research. Symbolic music has a rich and ambiguous structure, and the current research in symbolic music can shed light on learning hierarchical structure in natural language, image, as well as in art and real-world long-term planning in general.

---

> ### Author Response · Authors · 2023-11-19
> **Response to Reviewer JZXS (part two)**
>
> ### **Questions**
>
> **Q1: The diffusion model used auto regressively**
>
> Autoregressively applying diffusion model is indeed not a standard way. The method is the same as image inpainting and is first introduced in Polyffusion [2] as a way to generate coherent sequences. Such method is necessary for whole-song generation because as we generate lower level languages, the sequence becomes longer.
>
> The autoregressive generation process is introduced in Algorithm 1. Here we try to explain it more intuitively. For example, a diffusion model can generate a sequence of fixed length T, equal to the image width.  In the autoregressive process, the model first generates [0, T]. Then the model shifts right by T // 2 and inpainst [T, T + T//2] conditioning on the generated part [T//2, T], and so on so forth.
>
> **Q2: What are the pretrained models for chord progression, rhythmic pattern, accompaniment texture?**
>
> For both latent codes of chord progression and accompaniment texture, we use the pre-trained encoders in [4]. For latent code of rhythmic pattern, we use the pre-trained encoder in [5]. These are introduced in A.4 and we will make it clearer in our revision.
>
> **Q3: "Conterpoint" term for the second stage is not appropriate as this stage describes the sketch of the melody and harmony t. "Draft" or "Sketch" stage may be more suitable?**
>
> Thank you for the suggestion. We realize that the “Counterpoint” might not be precise. We will consider to use “Draft” or “Sketch”. And please let us know if you have other idea since we want the name to be more specific.
>
> In fact, “Counterpoint” is designed to emphasize the idea that the intermediate level music flow is represented by the interplay of melodic and harmonic development. From a Schenkerian perspective, we can regard melody and bass of the chord as a two-part counterpoint and the chord notes as “figured bass”.  This is what we mean by “Counterpoint” here.
>
> **Q4: Leadsheet encoded as melody and chords? Are chords in string format?**
>
> The *Lead sheet* is encoded as melody and chords. The chords are represented as piano-roll as discussed in section 3.2. Section A.2 shows an example of such data representation in piano-roll format.
>
> **Q5: What is the sampling time for a whole song?**
>
> We train and inference our model on GeForce RTX 2080 Ti. It takes several minutes (5 - 20 minutes) to generate a song depending on the length. For example, it takes approximately 15 minutes to generate eight pieces (batch size=8) of 40-measure songs. On more powerful computing resources, we expect better performance.
>
> ### **References**
>
> [1]  Ho J, Jain A, Abbeel P. Denoising diffusion probabilistic models. Advances in neural information processing systems. 2020;33:6840-51.
>
> [2] Lejun Min, Junyan Jiang, Gus Xia, Jingwei Zhao: Polyffusion: A Diffusion Model for Polyphonic Score Generation with Internal and External Controls. ISMIR 2023.
>
> [3] LeCun Y. A path towards autonomous machine intelligence version 0.9. 2, 2022-06-27. Open Review. 2022 Jun 27;62.
>
> [4] Ziyu Wang, Dingsu Wang, Yixiao Zhang, Gus Xia: Learning Interpretable Representation for Controllable Polyphonic Music Generation . ISMIR 2020: 662-669
>
> [5] Ruihan Yang, Dingsu Wang, Ziyu Wang, Tianyao Chen, Junyan Jiang, Gus Xia: Deep Music Analogy Via Latent Representation Disentanglement. ISMIR 2019: 596-603

---

### Official Review · Reviewer_QD5q · 2023-11-04

**Soundness:** 3 good
**Presentation:** 3 good
**Contribution:** 3 good
**Rating:** 8
**Confidence:** 4

**Summary:**

This paper proposed a new symbolic music generation system that can generate a full pop song. By defining four levels of music representations, the system generates piano roll from coarse to fine using a cascade diffusion model. The system is trained on POP909 and evaluated using objective metrics and a subjective listening test. The objective metrics show that the system can generate music with better long-term structural consistence. The listening test results show that the proposed system generates music of better quality.

**Strengths:**

- This paper is a timely and significant contribution to the field of symbolic music generation. As far as I know, this is the first deep music generation model that can generate a full structural song given high-level structural hints.
- The demo is impressive! Some of the samples are too good that I wonder if there is some overfitting issue (some nearest neighbor analysis might help clear these doubts).
- The proposed model is clever. Figure 4 clearly shows how the musical compositional hierarchy imposes a proper inductive bias to the system.

**Weaknesses:**

- A discussion on the limitations of the proposed system is missing. I see two main limitations: First, the musical compositional hierarchy adopted here is constrained to pop music. Second, the high-level form and structures still need to be provided.
- One thing missing in the evaluation is some nearest neighbor analysis to check if the model is returning part of the training data directly.
- The evaluation doesn't really measure the capability of whole-song generation, but there is no proper baselines as far as I know, so it's fine.

**Questions:**

- (Section 1) "and therefore we need to organize various music representations in a structured way." -> I cannot understand this sentence. Why do you mean by "organize representations"?
- (Section 3.2) "continuous" -> I'm not sure what "continuous" means here. Are you using binary or real-valued piano rolls?
- (Section 3.2) "Both melody reduction and simplified chord progression ..." -> How were these achieved? A pointer to the Appendix would be helpful here.
- (Section 3.2) "13 times" -> Why 13 times? Isn't it 128/12 = 11 times?
- (Section 3.3) "We select Sk relevant music segments
prior to t based on a defined similarity metric on X<k." -> Is this X^k or X^<k? The descriptions in this paragraph are somewhat confusing. From Figure 1, it seems like it's both  X^k and X^<k. Please clarify this.
- (Algorithm 1) Isn't the song length M also an input?
- (Section 4) "40 measures" -> How do you determine the number of measures to be generated?
- (Section 5.1) "... and segment them into 8-bar musical segments with a 1-bar hopping size." -> What is this segmentation step for?
- (Section 5.3) "Using pre-trained VAEs from Yang et al. (2019) and Wang et al. (2020)." -> Are these the models you used to extract autoregressive controls?
- (Section 5.3) How did you select the test inputs for the evaluation?

Here are some other comments and suggestions:

- (Section 1) "(typically ranging from a measure up to a phrase)" -> I don't think "phrase" has a strict definition.
- (Section 2.3) It's good to scope this section properly as only symbolic music models are discussed here.
- (Section 3.1) The musical compositional hierarchy discussed in this paper seems to be constrained to pop music. It would be help to discuss this limitation somewhere in the main text.
- (Section 3.1) "counterpoint" -> I personally find this term confusing as we have melody and harmony here -- it sounds more like a "reduced/simplified lead sheet" to me.
- (Section 3.3) "The time scopes (image widths) of these diffusion models are more or less the same" -> What do you mean by "more or less" the same? Please avoid such wordings.
- (Section 4) Having this qualitative analysis before introducing the dataset is somewhat misleading. I don't even know what I should expect. Please consider rearranging the sections.
- (Section 4) "In Appendix A.4, ..." -> I would love to see Figure 4 in the main text rather than Figure 2. The sheet music might be hard to understand for ICLR readers.
- (Section 4) "long (32 measures)" -> This is still far from whole song evaluation.
- (Section 5.3) "whole-song (32 measures)" -> I think 32 measures is still far from whole song generation.
- (Figure 3) It would be great to also include the ground truth for 32-measure generation.

**Details Of Ethics Concerns:**

~Discussions on the copyright of the dataset is missing. Also, a nearest neighbor analysis is missing, which is important to check if the model is returning part of the training data directly.~ -> addressed in the revision

---

> ### Author Response · Authors · 2023-11-19
> **Response to Reviewer QD5q (Part 1)**
>
> Thank you for your careful review and very detailed feedback. In this rebuttal, we first respond to the weaknesses (in part one) and then answer each of the questions (in part one and two).
>
> ### **Weakness 1: discussion of limitation is missing.**
>
> Thank you for pointing out our current limitation. We admit the current work is limited to pop music and we will put it in our revision.
>
> On the other hand, high-level form does not need to be given. In the training of stage one, the samples of *Form* are zero-padded to 256 measures. So in inference, the model can generate *Form* containing 256 measures and we derive the song length M by finding the first time step that contains all-zero entries. In section 4, we show music piece of 40 measures under given form for better visualization and space-saving purpose.
>
> ### **Weakness 2: Evaluation of overfitting.**
>
> We indeed see the evaluation of overfitting is crucial. We are conducting a systematic evaluation on the issue. Please see the **response to all reviewers** for more detail. In short, the current results show that our model does not suffer from plagiarism problem.
>
> ### **Weakness 3: No evaluation method of whole-song generation.**
>
> Directly evaluating whole-song generation is hard because 1) the evaluation of longer segments is more subjective; and 2) long music segments take a longer time to listen to and is hard to be memorized. In this paper, we propose to effectively evaluate whole-song generation by decomposing the evaluation problem into structure and quality evaluation:
>
> - Structure: a good structure usually implies clear phrase boundary, similarity between, e.g., two verses, and dissimilarity between, e.g., verse and chorus. These are evaluated by ILS in objective evaluation (Table 2) and “Boundary clarity” & “Phrase similarity” in subjective evaluation (Figure 3c&d).
> - Quality: we consider music quality in both short-term (8 measures) and long-term (32 measures) using criteria “Naturalness”, “Creativity” and “Overall musicality” (see Figure 3).
>
> Also, in the evaluation, whole songs are generated under several condensed music form such that the length of the piece is balanced: on one hand, the generated pieces won’t be too long (i.e., within 1’30’’); on the other hand, the music contains all necessary sections with repetition of verses or choruses.
>
> ### **Questions**
>
> - **(Section 1) "and therefore we need to organize various music representations in a structured way." -> I cannot understand this sentence. Why do you mean by "organize representations"?**
>
>     The common sense of music hierarchies are ambiguous. There are a lot of possible music hierarchies (e.g., pitch contour, tonal structure or metrical structure), but these hierarchies are usually not compatible with one another and do not serve for generation purpose. In this paper, we rely on these concepts to define music compositional hierarchy that can directly used in music generation.
>
> - **(Section 3.2) "continuous" -> I'm not sure what "continuous" means here. Are you using binary or real-valued piano rolls?**
>
>     The data are represented as real-valued piano-rolls and the generated real-valued piano-rolls will be quantized. The process is the same as Polyffusion [1], where the authors show such quantization operation has almost zero syntax error in practice.
>
> - **(Section 3.2) "Both melody reduction and simplified chord progression ..." -> How were these achieved? A pointer to the Appendix would be helpful here.**
>
>     These are computed using Tonal Reduction Algorithm discussed in A. 1. We will add the pointer here.
>
> - **(Section 3.2) "13 times" -> Why 13 times? Isn't it 128/12 = 11 times?**
>
>     It should be 11 times. We will fix the typo in our revision.
>
> - **(Section 3.3) "We select Sk relevant music segments prior to t based on a defined similarity metric on X<k." -> Is this X^k or X^<k? The descriptions in this paragraph are somewhat confusing. From Figure 1, it seems like it's both X^k and X^<k. Please clarify this.**
>
>     Sorry for the confusion. The term $X^{<k}$ here is correct. We use $X^{<k}$ as the criteria to select segments but embed the corresponding $X^{\leq k}$ as condition. This is not clearly addressed in the submission and we will discuss the procedure in detail in our revision.
>
> - **(Algorithm 1) Isn't the song length M also an input?**
>
>     No, it’s not. In the quantization step discussed above, we quantize the generated real-valued piano-rolls. In the training of stage one, the samples of *Form* are zero-padded to 256 measures. So in inference, the model outputs *Form* containing 256 measures and length M is determined by finding the first time step that contains all-zero entries. We will make it clearer in the revision.

---

> ### Author Response · Authors · 2023-11-19
> **Response to Reviewer QD5q (Part 2)**
>
> - **(Section 4) "40 measures" -> How do you determine the number of measures to be generated?**
>
>     Here the first-level language *Form* is given and so the length of the piece is given. We select a given 40-measure *Form* for better illustration and space-saving purpose.
>
> - **(Section 5.1) "... and segment them into 8-bar musical segments with a 1-bar hopping size." -> What is this segmentation step for?**
>
>     We apologize for the unclarity. It should be “We segment pieces into 32-bar segments to train the second stage of the model, and segment pieces into 8-bar segments to train the third and forth stage of the model”. We will revise the paper accordingly.
>
> - **(Section 5.3) "Using pre-trained VAEs from Yang et al. (2019) and Wang et al. (2020)." -> Are these the models you used to extract autoregressive controls?**
>
>     No. Unlike external condition, for autoregressive condition, we use a trainable encoder consisting of convolutional layers. We will make these details clearer in the revision.
>
> - **(Section 5.3) How did you select the test inputs for the evaluation?**
>
>     All the samples are randomly selected. Samples that have obvious failure will be discarded for both proposed and baseline methods. Such failure happens rarely, e.g., blank for one measure). For 8-measure subjective evaluation, the first two measures are given, and for 32-measure subjective evaluation, the first 4 measures are given. This setting reduce the possible variance in unconditional generation evaluation.
>
>     The reviewer also asks why not include ground truth samples in 32-measure subjective evaluation. The reason is that we regard 32-measure as a condensed pop song form (see response to Weakness 3). Such ground truth does not exist in the dataset. In the future, we should consider rewriting a ground truth piece into shorter piece for comparison.
>
>
> We are thankful to the reviewer to leave other comments and suggestions regarding organization of the paper and detailed wording choice. We agree with the reviewer in general and will consider revising the paper accordingly. (Note that some of the comments are addressed in detail in the above paragraphs.)
>
> Finally, as a response to the ethics flag, the arrangements in the POP909 dataset (the MIDIs and not the original audio) do not have copyright issue because it is made to the public by the authors of the POP909. Also, with our evaluation of plagiarism, we conclude that the plagiarism in our generation is unlikely, and if it happens, it can be detected.
>
> ### **References**
>
> [1] Lejun Min, Junyan Jiang, Gus Xia, Jingwei Zhao: Polyffusion: A Diffusion Model for Polyphonic Score Generation with Internal and External Controls. ISMIR 2023.

---

> > ### Comment · Reviewer_QD5q · 2023-11-22
> > **Response to the rebuttal**
> >
> > Thank you for the detailed rebuttal. The revisions look good to me. I've increased my score to 8.
> >
> > - Re song length M, thank you for the clarification. The added explanation in Section is helpful.
> > - Re overfitting and plagiarism concerns, the new experiments and the added appendix are great.
> > - Re the term "continuous" in Section 3.2, then maybe just use "real-valued" for clarity? (or something like "e.g., real-valued") Also, it's still unclear to me if you consider "velocity" or it's just relaxing {0, 1} to [0, 1].
> > - Re "For 8-measure subjective evaluation, the first two measures are given, and for 32-measure subjective evaluation, the first 4 measures are given.", this information of your experimental setup is important. Please add this to Section 5.3.

---

> > > ### Author Response · Authors · 2023-11-22
> > > **Response to Reviewer QD5q**
> > >
> > > Dear Reviewer QD5q,
> > >
> > > Thank you for your reply and acknowledging the changes we made are effective. We will make the other changes you mentioned in our next revision accordingly.

---

### Author Response · Authors · 2023-11-19
**Response to all reviewers (with plagiarism evaluation)**

We thank all the reviewers for the careful review and feedback. In this response to all reviewers, we would like to address the concern of generation plagiarism (i.e., copying the training set). This is raised by more than one reviewers and we see the question very important.

We conducted corresponding experiments to evaluate the degree of plagiarism. The current results show that our model does not suffer from plagiarism problem. The experiment is ongoing and we show some of the experimental results here. Later, we will summarize the results as a new section in the Appendix and will keep you posted.

### **Method**

Our evaluation method considers plagiarism in terms of melody similarity, a most recognizable aspect of music composition. Specifically, we consider a two-measure music segment $x$ and compute the similarity between the segment $x$ and $x’$, which refers to an arbitrary 2-measure segments in the training set. Then, we derive $s(x)$, being the largest possible similarity, i.e.,

$s(x) = \max_{x’ \in \text{Train}}\{\text{sim}(x, x’)\}$

Here, $s(x)$ is a value from 0 to 1 describing the property of $x$. A larger value of $s(x)$ show a higher degree of plagiarism. We can then measure the degree of plagiarism of a piece or a set of pieces by showing the distribution of $s(x)$.

Currently, we implemented a rule-based similarity metric to match exact copy of pitch classes and onsets. Formally, we compute the number of notes $n_{x=x’}$ that have the same pitch classes and onsets in segments $x$ and $x’$.  let $n_{x}$ and $n_{x’}$ be the number of notes in $x$ and $x’$, respectively. The similarity is defined to be the harmonic mean of $\frac{n_{x=x’}}{n_x}$ and $\frac{n_{x=x’}}{n_{x’}}$. **Note that we transpose the samples $x’$ in the training set to 12 keys to consider relative pitch similarity.**

### **Results**

Experiments show that

1. **(Test set)** The  average similarity between the test set and the training set is **0.6567 +/- 0.1141**.
2. **(Training set, Trivial)** The  average similarity between the training set and the training set itself is **1.0 +/- 0.0**.
3. **(Our proposed)** We randomly sample 128 pieces using our proposed algorithm. The average similarity between generated samples and the training set is **0.6530 +/- 0.1321**.
4. **(Baseline)** We implement a baseline “copybot” which always copies the training set (each measure copies different part of the training set). The average similarity between samples generated by the copybot and the training set is **0.7108 +/- 0.1159.**

The results show that our model does not suffer from plagiarism problem.

Moreover, we also test the similarity of generated samples/demos in the submitted paper and the website. We show that all samples are original (except one sample, the third sample in section “More Examples of Whole-song Generation”).

### **Next step**

We are working on more flexible methods to compare melody similarity (e.g., using a pre-trained VAE and measure the cosine similarity of latent codes). We will share with you the results later. Also, we will let you know when the appendix section on this issue is ready.

---

### Author Response · Authors · 2023-11-21
**Paper revision uploaded**

Dear reviewers,

Thank you for your comments to improve this paper. We have made a revision and here is a summary of changes:
1. We added a section on generation plagiarism in Appendix A.5.
2. We added more details about our model architecture and training detail (in section 3.3 and A.3).
3. We added several sentences in section 4 to better explain our evaluation method of whole-song generation.
4. We make the letter "s" in the title "Whole-Song Generation" capitalized to follow the style guides.
5. Other minor changes.

**All the changes are highlighted in blue.** Other changes suggested by the reviewers will be added to the paper shortly afterwards.

---

### Meta-Review · Area_Chair_yC5K · 2023-12-06

**Metareview:**

This paper describes a novel symbolic music generation system that can generate a full pop song. Using four discrete levels of music representations, the system generates piano roll from coarse to fine using a cascade diffusion model. The system is trained on (the small-ish) POP909 database and evaluated using objective metrics and a subjective listening test. The objective metrics show that the system can generate music with better long-term structural consistence. The listening test results show that the proposed system generates music of better quality. Discussion confirmed that the model does indeed generate novel music (and doesn't just reproduce training data).

Strengths:
- This paper is a timely and significant contribution to the field of symbolic music generation, which has a long history at ICLR. This seems to be the first deep music generation model that can generate a full structural song given high-level structural hints.
- The demo is impressive and authors addressed the training data reproduction issue
- The model is intuitive and well implemented, the musical compositional hierarchy imposes a proper inductive bias to the system.

Weaknesses:
- Such a model is difficult to evaluate, but reviewers agree after discussion that the authors have addressed their concerns.
- A discussion (and study) on the limitations of the proposed system is missing, i.e. is this approach limited to pop music? Which other high-level forms and structures can be supported?
- Somewhat of a "niche" topic, but very well executed, and aspects such as the hierarchical modeling may help in other domains as well.
- If published, authors need to re-word the "plagiarism" discussion -- the model copying training data is not "plagiarism", but just that - the model copying training data.

**Justification For Why Not Higher Score:**

This is a niche topic (although aspects such as hierarchical modeling may be relevant to other areas as well), and reviewers have still some reservations about the quality (reliability) of results, therefore "oral" seems risky

**Justification For Why Not Lower Score:**

The paper is better and the presentation is more polished than most "poster" papers. Reviewers are genuinely excited about the presented material, and good demo examples, which would make for a good spotlight presentation.

---

### Decision · Program_Chairs · 2024-01-16

Accept (spotlight)